# Water-dispersible X-ray scintillators enabling coating and blending with polymer materials for multiple applications

Hailei Zhang [1,2] ✉, Bo Zhang[1], Chongyang Cai[3], Kaiming Zhang [2], Yu Wang[1], Yuan Wang[1], Yanmin Yang[3] ✉, Yonggang Wu [1], Xinwu Ba[1] & Richard Hoogenboom [2] ✉

Developing X-ray scintillators that are water-dispersible, compatible with polymeric matrices, and processable to flexible substrates is an important challenge. Herein, $Tb^{3+}$-doped $Na_5Lu_9F_{32}$ is introduced as an X-ray scintillating material with steady-state X-ray light yields of 15,800 photons $MeV^{-1}$, which is generated as nanocrystals on halloysite nanotubes. The obtained product exhibits good water-dispersibility and highly sensitive luminescence to X-rays. It is deposited onto a polyurethane foam to afford a composite foam material with dose-dependent radioluminescence. Moreover, the product is dispersed into polymer matrixes in aqueous solution to prepare rigid or flexible scintillator screen for X-ray imaging. As a third example, it is incorporated multilayer hydrogels for information camouflage and multilevel encryption. Encrypted information can be recognized only by X-ray irradiation, while the false information is read out under UV light. Altogether, we demonstrate that the water-dispersible scintillators are highly promising for aqueous processing of radioluminescent, X-ray imaging, and information encrypting materials.

X-ray scintillators have emerged as excellent light emitting radioluminescent materials, which can convert high energy X-ray radiations into visible or near visible light that may be further captured and converted into visual information or electrical signals by a photomultiplier[1–3]. Such attractive characteristics make X-ray scintillators promising materials for applications in the fields of X-ray detectors[4], space exploration[5], medical imaging[6], light-emitting diodes[7], and radiation exposure monitoring[8]. Straightforward incorporation of X-ray scintillators into polymeric materials would represent a huge breakthrough in organic-inorganic composite materials, which is quite beneficial to mount multi-components[9], satisfy the requirement of flexible screens[10], and extend their application in a wider range of fields[11,12]. Despite many kinds of X-ray scintillators have been reported in past decades[13,14] and the light yield has been

improved to a great extent (Supplementary Tables 1 and 2), developing X-ray scintillators as polymer composite materials is still full of difficulties and challenges. Even though some single crystals exhibit high light yields (Supplementary Table 1), the harsh growth conditions, nonflexibility, and high fabrication cost limited their application to conventional hard devices. The poor water-dispersibility of X-ray scintillating crystals and particles also results in limited processability in water. Moreover, aggregation of inorganic particles is commonly observed leading to microphase separation and inhomogeneous composites with compromised performance. The toxicity of lead and unpredicted thermal quenching effects of perovskite-based X-ray scintillators (Supplementary Table 2) are remaining challenges[15–18]. Organic X-ray scintillators have also been developed in recent research[19], but usually suffer from a limited effective atomic number[20]

[1]College of Chemistry & Materials Science, Hebei University, 180 Wusi Road, 071002 Baoding, China. [2]Supramolecular Chemistry Group, Centre of Macromolecular Chemistry (CMaC), Department of Organic and Macromolecular Chemistry, Ghent University, Krijgslaan, 281-S4, 9000 Gent, Belgium. [3]College of Physics Science and Technology, Hebei University, 180 Wusi Road, 071002 Baoding, China. ✉e-mail: zhanghailei@hbu.edu.cn; mihuyym@163.com; richard.hoogenboom@ugent.be

and inefficient excitation efficiency[21]. Therefore, the development of X-ray scintillator materials that are water-dispersible, low-toxic, highly sensitive to X-rays under room temperature, compatible with polymer matrices, and processable to flexible substrates remains an unmet challenge[22,23].

Halloysite nanotubes (HNTs), with the chemical composition of $Al_2Si_2O_5(OH)_4 \cdot n H_2O$, are a naturally occurring tubular product and have raised people's attention in recent years[24,25]. The large cavity results in deduced density, which is beneficial for achieving good dispersibility in water[26,27]. The HNTs exhibit different surface charge on the outer (Si-O-Si) and inner (Al-OH) surfaces[28], whereby the stable negative charge on the outer surface can prevent aggregation of the nanotubes[29]. Other attractive properties of HNTs include their good biocompatibility[30,31], low toxicity[32], high stability[33], hydrophilicity[34], processability[35], and low-cost[36] making HNTs promising candidates for the fabricating of composite materials. Though there is still some controversy on the potential toxicity upon some special items[37], HNTs remain a popular topic in material science including biomedical uses with a growing trend of attention[38]. The existence of silanol groups on the surface allows straightforward surface functionalization[36,39]. These favourable properties of HNTs inspired us to explore them as a substrate to prepare lead-free nanocomposites with X-ray scintillating abilities, which may have good water-dispersibility and desirable compatibility in polymeric matrices. Though numerous methods for surface modification of nanoscintillators have already been reported, anchoring nanoscintillators on HNTs with large length-diameter ratios is expected to improve the tensile strength of X-ray-sensitive composite materials[40], which is quite important for emerging flexible scintillator devices[41,42].

In this work, we aim to develop water-dispersible X-ray scintillators based on HNTs modified with lead-free X-ray scintillator nanoparticles. $Tb^{3+}$ has been demonstrated to be a promising doping ion in luminescent crystals. Liu et al. synthesized a series of $Tb^{3+}$-doped $NaLuF_4$ nanoparticles which can emit more than 30 days of persistent radioluminescence[43]. Another kind of nanocrystal, $Tb^{3+}$-doped $Na_5Lu_9F_{32}$ ($Na_5Lu_9F_{32}{:}Tb^{3+}$), bearing the same elements in different ratios has also been reported[44], but their X-ray scintillator properties have not been reported. Fortunately, we discovered that the $Na_5Lu_9F_{32}{:}Tb^{3+}$ nanocrystals can emit strong green luminescence upon exposure to X-ray irradiation and, hence, exhibit X-ray scintillator behavior (Supplementary Fig. 1). In contrast to the persistent radioluminescence from $NaLuF_4{:}Tb^{3+}$ (Supplementary Fig. 2), $Na_5Lu_9F_{32}{:}Tb^{3+}$ shows a synchronous radioluminescence (RL) behavior following the on-off switching of X-ray irradiation and is suitable to be applied in the fields of X-ray monitoring, imaging, and other X-ray-sensitive materials. To generate $Na_5Lu_9F_{32}{:}Tb^{3+}$ on the surface of the HNTs, the outer surface is modified with citric acid (CA) as a chelating agent by a typical condensation reaction with the amino groups of

aminated HNTs (HNTs-$NH_2$). The CA modification of the HNTs directs the nanoparticle crystal growth onto the surface rather than in the medium. Following this way, $Na_5Lu_9F_{32}{:}Tb^{3+}$ nanocrstyals are successfully anchored on the surface of the nanotube with well-controlled particle size to afford the obtained product $Na_5Lu_9F_{32}{:}Tb^{3+}$-anchored halloysite nanotubes (HNTs@$Na_5Lu_9F_{32}{:}Tb^{3+}$) with X-ray scintillating abilities, good dispersibility (Supplementary Fig. 3), and desirable compatible with the polymeric matrices as will be demonstrated in this work. Furthermore, three kinds of materials are developed targeting different applications, including HNTs@$Na_5Lu_9F_{32}{:}Tb^{3+}$ adsorption onto the surface of polyurethane foam (PUF) resulting in a composite foam with radioluminescent properties that may be utilized for radiation exposure monitoring, HNTs@$Na_5Lu_9F_{32}{:}Tb^{3+}$ dispersion into a carboxymethylcellulose sodium (CMC-Na) solution or chemically-crosslinked hydrogel to prepare a rigid or flexible scintillator screen for X-ray imaging, HNTs@$Na_5Lu_9F_{32}{:}Tb^{3+}$ incorporation into hydrogels to fabricate multilayer hydrogels that can be used for information encryption. True information can only be identified under X-rays, while false information can be detected when exposed to UV light, which has the potential for information encryption and personal recognition.

## Results and discussion
### Synthesis and characterization
CA is a commonly-used chelating agent to fabricate nanocrystal scintillators[45]. In our strategy, CA is coupled to the outer surface of HNTs-$NH_2$ (Fig. 1). The obtained citric acid-modified halloysite nanotubes (HNTs-CA) are thoroughly washed to ensure the absence of free CA, which is an important precondition for generating the X-ray scintillator on the surface of nanotubes. The HNTs-CA are found to have good dispersibility in water and are favored by solid-state $^{13}C$ NMR, FTIR, and TGA (Supplementary Figs. 4–6). Detailed results and discussion on the synthesis of HNTs-CA are included in the Supplementary Discussion 1.

After confirming the successful preparation of the HNTs-CA, they are used to template the growth of the X-ray scintillator nanoparticles using a hydrothermal reactor to prepare HNTs@$Na_5Lu_9F_{32}{:}Tb^{3+}$ as depicted in Fig. 1. Therefore, the HNTs-CA are added to an aqueous solution of $NaNO_3$, $Lu(NO_3)_3 \cdot 6H_2O$, and $Tb(NO_3)_3 \cdot 6H_2O$ with the molar ratio of 6 : 10 : 1. Subsequently, $NH_4F$ is added and the medium is slightly acidified with nitric acid to pH 5, which is a crucial factor to prepare the nanocomposite, followed by hydrothermal treatment at 180 °C. A neutral environment can give rise to the formation of $NaLuF_4{:}Tb^{3+}$, rather than $Na_5Lu_9F_{32}{:}Tb^{3+}$.

The mixture of $NaNO_3$, $NH_4F$, and $Lu(NO_3)_3 \cdot 6H_2O$ can afford the $Na_5Lu_9F_{32}$ as luminescent host materials, while the $Tb^{3+}$ served as an activator that is excited upon X-ray irradiation. The as-obtained product, HNTs@$Na_5Lu_9F_{32}{:}Tb^{3+}$, is carefully characterized by XPS to reveal the chemical compositions.

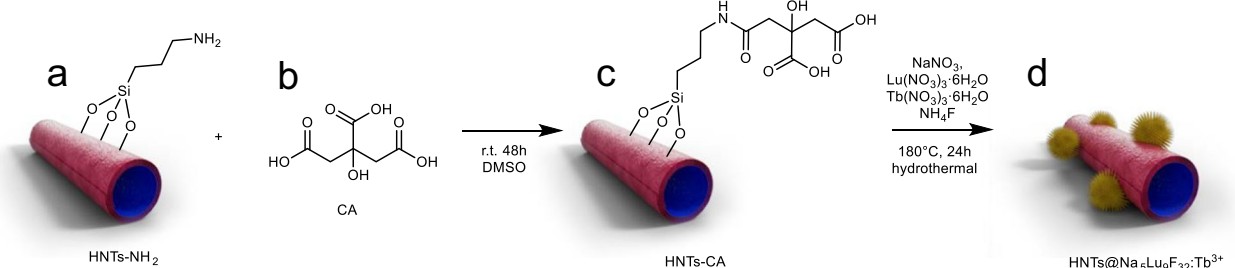

**Fig. 1 | Synthesis of water-dispersible X-ray scintillator. a** Aminated halloysite nanotubes (HNTs-$NH_2$). **b** Chemical structure of citric acid (CA). **c** Citric acid-modified halloysite nanotubes (HNTs-CA). HNTs-CA is synthesized by treating HNTs-$NH_2$ with CA, 1-(3-dimethylaminopropyl)−3-ethylcarbodiimide hydrochloride, 1-hydroxybenzotriazole, and N,N-diisopropylethylamine in dimethylsulfoxide (DMSO) at room temperature (r.t.) for 48 h. **d** $Tb^{3+}$-doped $Na_5Lu_9F_{32}$ anchored halloysite nanotubes (HNTs@$Na_5Lu_9F_{32}{:}Tb^{3+}$). HNTs@$Na_5Lu_9F_{32}{:}Tb^{3+}$ is synthesized by placing HNTs-CA, $NaNO_3$, $Lu(NO_3)_3 \cdot 6H_2O$, $Tb(NO_3)_3 \cdot 6H_2O$, nitric acid, $NH_4F$, and water in a hydrothermal reactor with a filling rate of 70% and then heated at 180 °C for 24 h.

Silicon, aluminum, and oxygen are the main elements detected in the X-ray photoelectron spectroscopy (XPS) pattern of the HNTs@Na$_5$Lu$_9$F$_{32}$:Tb$^{3+}$ (Fig. 2). The Si 2 s, Al 2 s, Si 2p, and Al 2p peaks can be observed at 153, 118, 102, and 73 eV, respectively (Fig. 2a–c). Each silicon or aluminum atom in the halloysite is coordinated to 3 ~ 4 oxygen atoms. As a result, the O 1 s peak at a binding energy of 531 eV is significantly higher than the others (Fig. 2a). The intensity of the C 1 s peak significantly increases in the XPS pattern of HNTs@Na$_5$Lu$_9$F$_{32}$:Tb$^{3+}$ as compared to that of pristine HNTs[46]. A newly emerged N 1 s peak is observed at a binding energy of 400.9 eV corresponding to the amide nitrogens of the (modified) HNT-NH$_2$. To further characterize the composition, the regions relating to C 1 s are expanded as shown in Fig. 2d. The peaks at 284.8 eV and 289.0 eV should be attributed to the carbons in C–C bonds and C = O groups, respectively. The appearance of C = O groups confirms the presence of CA moieties and matches well with FTIR findings. The presence of sodium and fluorine is demonstrated by the Na 1 s and F 1 s peaks at 1,071.3 and 684.4 ppm, respectively (Fig. 2e, f). The existence of lutecium in the HNTs@Na$_5$Lu$_9$F$_{32}$:Tb$^{3+}$ is confirmed by the Lu 4d5 peak at binding energies of 206.3 and 196.7 eV (Fig. 2g). The doublet at 1,276.8 and 1,243.3 eV can be assigned to the Tb 3d signals (Fig. 2h), confirming the presence of terbium. The chemical composition resulting from XPS is in accordance with the expected product. Moreover, the presence of Na$_5$Lu$_9$F$_{32}$:Tb$^{3+}$ (JCPDF no. 27-0725) and halloysite (JCPDF no. 09-0453) in the obtained product can be evidenced by the powder X-ray diffractogram (PXRD) (Fig. 2i and Supplementary Fig. 7).

The micromorphology of the HNTs@Na$_5$Lu$_9$F$_{32}$:Tb$^{3+}$ is analyzed using scanning transmission electron microscopy (STEM) coupled with energy-dispersive X-ray spectroscopy (EDX). The pristine HNTs depicted in Supplementary Fig. 8 show a smooth external surface, while the obtained HNTs@Na$_5$Lu$_9$F$_{32}$:Tb$^{3+}$ displays distinct differences: some newly emerged nanoparticles generated from the nanotube's external surface can be observed (Fig. 2j, k). The high-magnification TEM image of HNTs@Na$_5$Lu$_9$F$_{32}$:Tb$^{3+}$ shown in Fig. 2l and XRD results suggest that the lattice of the attached nanoparticles is in accordance with the isometric system (a = b = c). The high-angle annular dark-field (HAADF) STEM images in Supplementary Fig. 9 shows an obvious contrast between the nanotubes and the attached particles, suggesting that the anchored particles consist of elements with higher atomic numbers than those of pristine HNTs (Si, Al, and O). Figure 2m–r shows the STEM-energy dispersive X-ray (STEM-EDX) elemental mapping of HNTs@Na$_5$Lu$_9$F$_{32}$:Tb$^{3+}$. The Al and Si plots can only be detected on the nanotubes, in agreement with the character of aluminosilicate. On the contrary, the plots from Na, Lu, F, and Tb show their distribution on the newly generated nanoparticles. The high coincidence of Tb with Na, Lu, and F also indicates that Tb should be effectively doped into the host lattices. The XPS, XRD, and TEM results indicate that Na$_5$Lu$_9$F$_{32}$:Tb$^{3+}$ nanoparticles are generated onto the nanotubes, and thereby the expected product HNTs@Na$_5$Lu$_9$F$_{32}$:Tb$^{3+}$ is obtained.

The emission properties of pristine HNTs, HNTs-CA, and HNTs@Na$_5$Lu$_9$F$_{32}$:Tb$^{3+}$ are investigated by X-ray irradiation (RL spectra, Fig. 3a), which yields completely different luminescence emission behaviors. The pristine HNTs and HNTs-CA exhibit no luminescence properties when exposed to X-ray irradiation. As for HNTs@Na$_5$Lu$_9$F$_{32}$:Tb$^{3+}$, the RL spectrum features four emission peaks at 489, 544, 585, and 620 nm, in which the intensity of the peak at 544 nm is much higher than the others. As a result, the HNTs@Na$_5$Lu$_9$F$_{32}$:Tb$^{3+}$ exhibit strong green luminescence when exposed to X-ray irradiation. The peak positions are exactly the same to the RL spectrum from Na$_5$Lu$_9$F$_{32}$:Tb$^{3+}$. The light yields of Na$_5$Lu$_9$F$_{32}$:Tb$^{3+}$ and HNTs@Na$_5$Lu$_9$F$_{32}$:Tb$^{3+}$ are calculated to be 15,800 and 12,300 photons MeV$^{-1}$, respectively (Supplementary Fig. 10). Then the sensitivity of HNTs@Na$_5$Lu$_9$F$_{32}$:Tb$^{3+}$ towards X-ray is evaluated based on on-off cycles (Fig. 3b). The HNTs@Na$_5$Lu$_9$F$_{32}$:Tb$^{3+}$ exhibits a synchronous RL behavior following the on-off switching of X-ray

irradiation. Importantly, the developed HNTs@Na$_5$Lu$_9$F$_{32}$:Tb$^{3+}$ can also be stable isolated in a dry state and redispersed in water for further processing (Supplementary Fig. 3).

## X-ray-sensitive polyurethane foam

PUF is a commonly-used and commercially available polymeric material, which is widely used in textile, furniture, and construction[47]. Hence, the ability to functionalize preformed PUF with X-ray scintillators would provide straightforward access to flexible polymer products with X-ray scintillating properties. Therefore, we hypothesize that it might be able to coat a PUF with an aqueous dispersion of the developed HNTs@Na$_5$Lu$_9$F$_{32}$:Tb$^{3+}$. For this purpose, the PUF sample is immersed in an aqueous dispersion of the HNTs@Na$_5$Lu$_9$F$_{32}$:Tb$^{3+}$ X-ray scintillator followed by removal of excess water via thorough compressing (Fig. 4). This procedure is repeated for three cycles to obtain a homogeneous coating. Benefiting from the presence of hydroxyl groups on the surface, the HNTs are easily adsorbed onto the polymeric matrix providing a good adhesion between HNTs@Na$_5$Lu$_9$F$_{32}$:Tb$^{3+}$ and the PUF, since very little powder peeled off from the HNTs@Na$_5$Lu$_9$F$_{32}$:Tb$^{3+}$ coated PUF (HNTs@Na$_5$Lu$_9$F$_{32}$:Tb$^{3+}$@PUF) during use. The soak time for adsorption is limited to 3 min because a longer time cannot afford better adhesion behavior as determined by a gravimetrical method (Supplementary Fig. 11).

The micromorphology of the pristine PUF and the HNTs@Na$_5$Lu$_9$F$_{32}$:Tb$^{3+}$@PUF is investigated by SEM (Fig. 5a–g). The PUF exhibits open-cell structures with smooth pore walls (Fig. 5a–c). The framework can also be observed without damage in the SEM image of HNTs@Na$_5$Lu$_9$F$_{32}$:Tb$^{3+}$@PUF shown in Fig. 5d. It should be noted that the surface of the pore walls turns to be rougher in Fig. 5e, f, indicative of successful adsorption of the HNTs@Na$_5$Lu$_9$F$_{32}$:Tb$^{3+}$ X-ray scintillators. In fact, the nanotubes can be observed in the SEM image under higher magnification times (Fig. 5g). SEM image in Fig. 5h is further analyzed by elemental mapping. Elemental mapping of HNTs@Na$_5$Lu$_9$F$_{32}$:Tb$^{3+}$@PUF indicates the distribution of Al, Si, Na, and F at the pore walls (Fig. 5i–l). Owing to the low content of Tb in the composite, the signal of Tb$^{3+}$ is much lower than the others (Fig. 5m). Nonetheless, the presence of Tb$^{3+}$ confirms the successful adsorption of the HNTs@Na$_5$Lu$_9$F$_{32}$:Tb$^{3+}$ as it matches well with elemental distributions of Al, Si, Na, and F present in the HNTs. To further demonstrate the uniformity of the coating, five sites in the obtained HNTs@Na$_5$Lu$_9$F$_{32}$:Tb$^{3+}$@PUF sample are randomly selected and the peak intensity at 544 nm in the RL spectra of each site is recorded. The relative standard deviation (RSD) value of the results from the intensity values is 4.8%, suggesting a uniform coating of HNTs@Na$_5$Lu$_9$F$_{32}$:Tb$^{3+}$ on the PUF surface. A good penetration ability of the obtained HNTs@Na$_5$Lu$_9$F$_{32}$:Tb$^{3+}$ into the pores of the PUF is demonstrated by elemental analysis on the cross-section (shown in Supplementary Fig. 12 and Supplementary Table. 3). The results of a 30 day stability test show that the obtained HNTs@Na$_5$Lu$_9$F$_{32}$:Tb$^{3+}$@PUF exhibits good stability to heat and good light stability (Supplementary Fig. 13).

When exposed to X-rays, the obtained HNTs@Na$_5$Lu$_9$F$_{32}$:Tb$^{3+}$@PUF (Supplementary Fig. 14a) can emit visible green light (Supplementary Fig. 14b–f), which matches well with the X-ray-excited RL emission spectrum. Moreover, the luminescence intensity enhances with the increase of the X-ray dose from 3.1 to 9.2 cGy s$^{-1}$. This developed aqueous coating of the HNTs@Na$_5$Lu$_9$F$_{32}$:Tb$^{3+}$ X-ray scintillators on PUF provides opportunities for fabrication of protective clothing, surgical gowns, or even a protective wall for an X-ray room, for in-situ radiation exposure monitoring.

To achieve this, the obtained HNTs@Na$_5$Lu$_9$F$_{32}$:Tb$^{3+}$@PUF samples (Fig. 5n) are cut to the required shapes (Fig. 5o) and then attached to a lab coat via typical sewing operations (Fig. 5p). We also incorporate the obtained HNTs@Na$_5$Lu$_9$F$_{32}$:Tb$^{3+}$ into an epoxy resin (Fig. 5q) and fabricate it into a drop-shaped pendant with good transparency

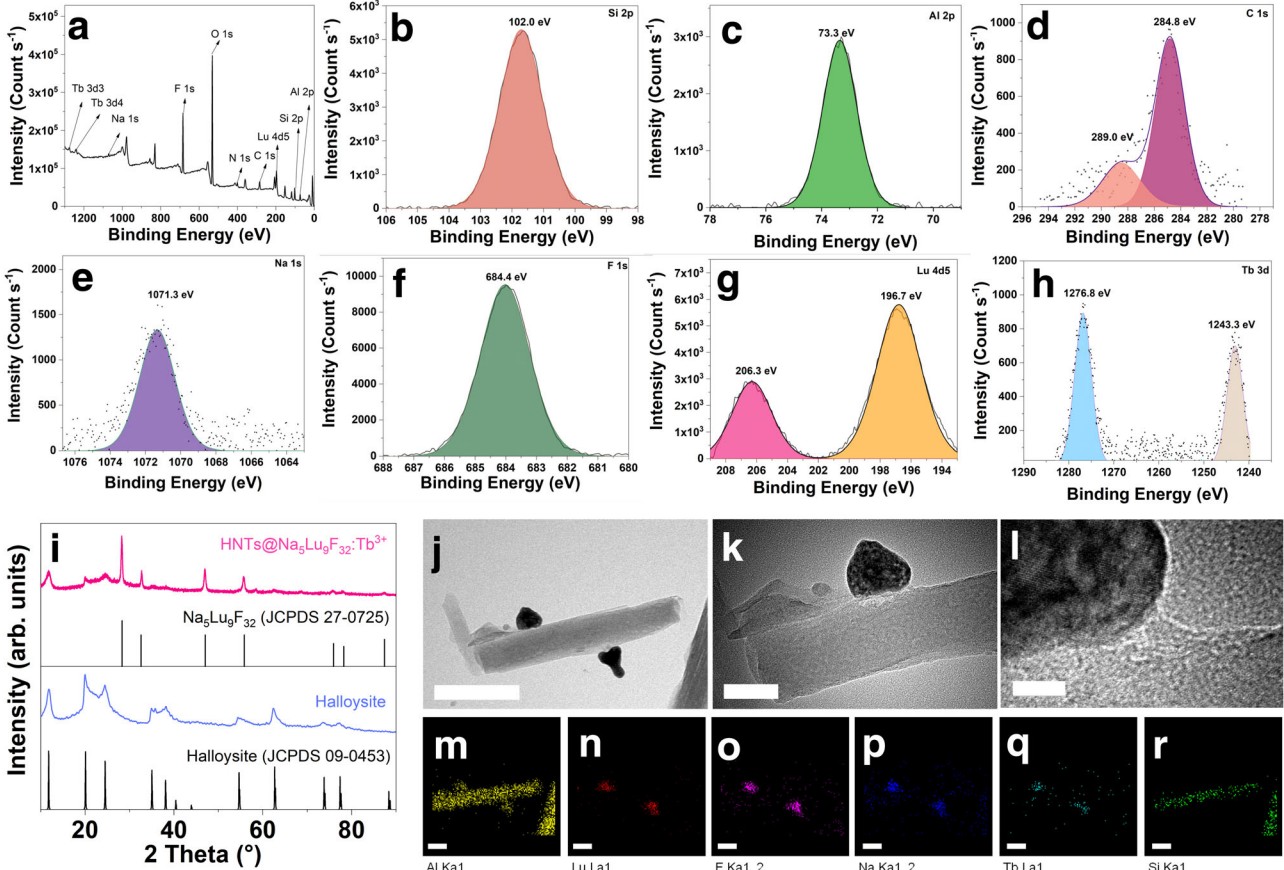

**Fig. 2 | Structural and micromorphology characterizations. a** X-ray photoelectron spectroscopy (XPS) pattern of $Tb^{3+}$-doped $Na_5Lu_9F_{32}$ anchored halloysite nanotubes (HNTs@$Na_5Lu_9F_{32}$:$Tb^{3+}$). **b** Si 2p region. **c** Al 2p region. **d** C 1s region. **e** Na 1s region. **f** F 1s region. **g** Lu 4d5 region. **h** Tb 3d region. **i** X-ray powder diffraction (XRD) pattern of halloysite, HNTs@$Na_5Lu_9F_{32}$:$Tb^{3+}$, and standards. **j** Transmission electron microscopy (TEM) image of HNTs@$Na_5Lu_9F_{32}$:$Tb^{3+}$ (bar: 200 nm). **k** TEM image of HNTs@$Na_5Lu_9F_{32}$:$Tb^{3+}$ (bar: 50 nm). **l** High resolution TEM image of HNTs@$Na_5Lu_9F_{32}$:$Tb^{3+}$ (bar: 10 nm). **m**–**r** Elemental mapping analysis of HNTs@$Na_5Lu_9F_{32}$:$Tb^{3+}$ (**m** Al. **n** Lu. **o** F. **p** Na. **q** Tb. **r** Si; bar: 100 nm). Source data in **a**-**i** are provided as Source Data files.

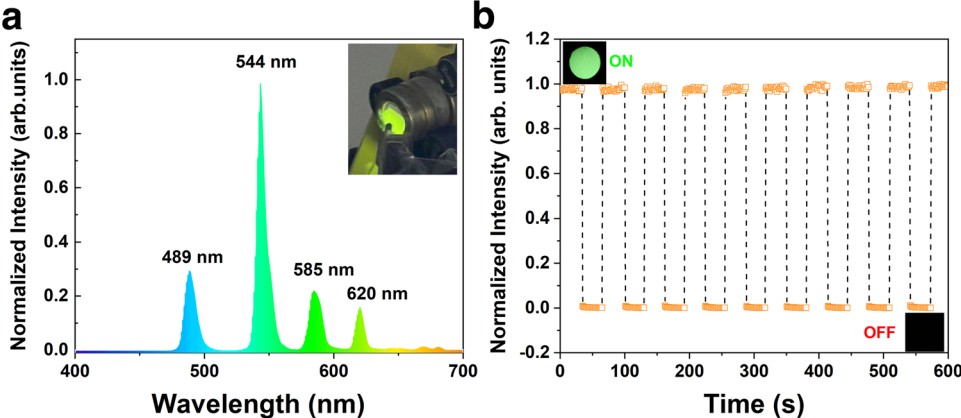

**Fig. 3 | Optical property. a** X-ray-excited radioluminescence (RL) spectrum of $Tb^{3+}$-doped $Na_5Lu_9F_{32}$ anchored halloysite nanotubes (HNTs@$Na_5Lu_9F_{32}$:$Tb^{3+}$) and the photographic image of HNTs@$Na_5Lu_9F_{32}$:$Tb^{3+}$ taken under X-ray. **b** In situ measurement of the luminescence intensity of HNTs@$Na_5Lu_9F_{32}$:$Tb^{3+}$ under X-ray with on-off cycles. Source data in (**a**) and (**b**) are provided as Source Data files.

(Fig. 5r). The obtained HNTs@$Na_5Lu_9F_{32}$:$Tb^{3+}$@PUF-bearing lab coat and drop shape pendant are dressed on a full-length mannequin. We apply the obtained materials in radiological laboratories. The results indicated that both the foams (Fig. 5s) and drop-shaped pendant (Fig. 5t) can emit visible green light under low dose X-ray (3.1 cGy s$^{-1}$) which is much lower than that used in brachytherapy[48,49].

## Rigid and flexible X-ray scintillator screens

Thin-layer chromatographical plates (TLCP) are commercially available and commonly used materials in the field of organic chemistry. Typically, TLCP is prepared by mixing silica gel or $Al_2O_3$ with CMC-Na followed by coating on a glass slide. As the surface of HNTs mainly consists of Si-O-Si and Al-OH groups, which is chemically similar to that

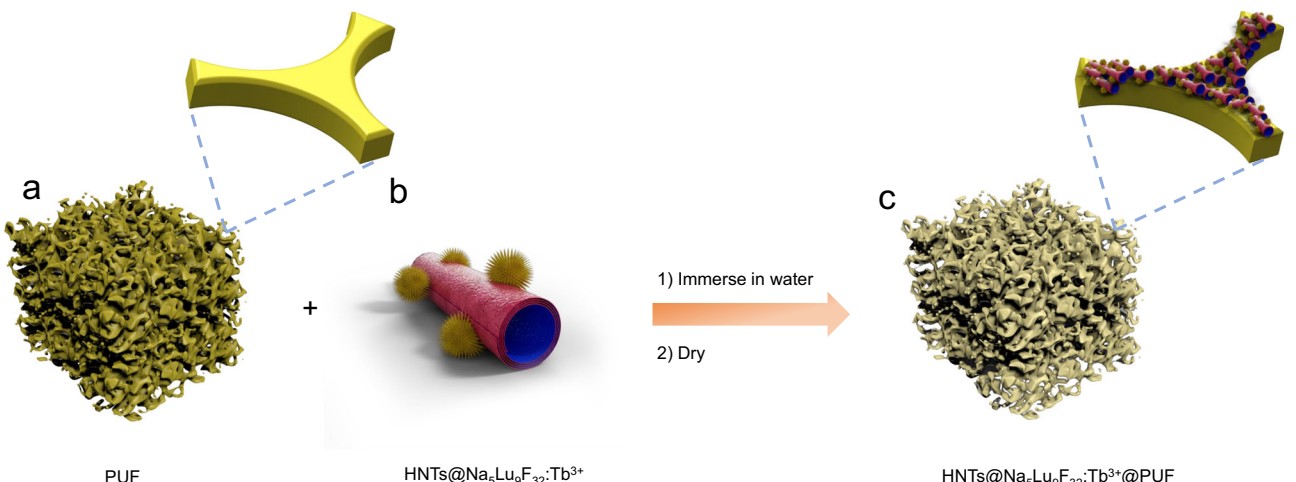

**Fig. 4 | Preparation of X-ray-sensitive polyurethane foam. a** Polyurethane foam (PUF); **b** $Tb^{3+}$-doped $Na_5Lu_9F_{32}$ anchored halloysite nanotubes ($HNTs@Na_5Lu_9F_{32}:Tb^{3+}$); **c** $Tb^{3+}$-doped $Na_5Lu_9F_{32}$ anchored halloysite nanotubes coated polyurethane foam ($HNTs@Na_5Lu_9F_{32}:Tb^{3+}@PUF$). $HNTs@Na_5Lu_9F_{32}:Tb^{3+}@PUF$ is prepared by immersing PUF in the $HNTs@Na_5Lu_9F_{32}:Tb^{3+}$ aqueous dispersion by fully compressing and soaking.

of $Al_2O_3$ and silica gel, respectively, it inspires us to develop a solid-state X-ray scintillator screen by mixing the obtained $HNTs@Na_5Lu_9F_{32}:Tb^{3+}$ with CMC-Na following a similar preparation procedure as used for production of TLCP (Fig. 6). The schematic diagram for the preparation of the X-ray scintillator screen is shown in Fig. 7a based on the coating of a glass plate with the CMC-Na solution containing the dispersed $HNTs@Na_5Lu_9F_{32}:Tb^{3+}$. The resulting TLCP-like solid-state X-ray scintillator screen is found to have a macroscopically smooth and uniform surface resulting from the good coating properties of the CMC-Na combined with the good water-dispersibility of the $HNTs@Na_5Lu_9F_{32}:Tb^{3+}$ (Fig. 7b), which provides strong green RL under low-dose X-ray irradiation.

The feasibility of X-ray imaging is investigated by using a capsule model (Fig. 7c) and a self-made imaging apparatus (Fig. 7d). A metal-based spring is loaded in a non-transparent capsule and the obtained capsule model is used as the test sample. The as-prepared solid-state X-ray scintillator screen is placed in front of the X-ray beam path (30 kV, 20 mA) and a digital camera is employed to record the graphic information on the screen. The inside content cannot be disclosed by the naked eye, while a clear image reflecting a spring-like form can be observed on the $HNTs@Na_5Lu_9F_{32}:Tb^{3+}$-based X-ray scintillator screen and can be recorded on the digital camera. Meanwhile, the outline of the capsule can also be monitored in the recorded image (Fig. 7e). Moreover, the stability test showed the obtained $HNTs@Na_5Lu_9F_{32}:Tb^{3+}$-based X-ray scintillator screen exhibits good stability to heat and good light stability (Supplementary Fig. 13). Even though similar high-resolution X-ray imaging is previously demonstrated using other X-ray scintillators, especially perovskite-based, under lower X-ray doses, the developed lead-free composite X-ray imaging film in this study can be prepared in a much simpler water-based procedure that is similar to the fabrication of TLCP, which is already performed in mass production indicating the scalability of the $HNTs@Na_5Lu_9F_{32}:Tb^{3+}$ based X-ray imaging screen.

To overcome the restrictions of rigid scintillator screens, we demonstrate a facile approach to prepare a flexible scintillator screen by incorporating $HNTs@Na_5Lu_9F_{32}:Tb^{3+}$ into a chemically crosslinked hydrogel (Supplementary Fig. 15). Flexible scintillator screens are favored to be used to accommodate non-flat objects by stretching or bending into required shapes[41,42]. Therefore, the tensile strength is a very important property for flexible scintillator screens and should be carefully evaluated. Additionally, we also made attempts to modify the surface of $Na_5Lu_9F_{32}:Tb^{3+}$ by oleic acid (OA) to give the product $OA@Na_5Lu_9F_{32}:Tb^{3+}$ (Supplementary Fig. 16) resulting in improved water dispersibility. The incorporation of $OA@Na_5Lu_9F_{32}:Tb^{3+}$ in the hydrogel is, however, found to lower the tensile strength compared to the pristine hydrogel. Anchoring of the $Na_5Lu_9F_{32}:Tb^{3+}$ on the surface of HNTs is found to overcome this. The $HNTs@Na_5Lu_9F_{32}:Tb^{3+}$-containing hydrogel shows a much better tensile strength than the $OA@Na_5Lu_9F_{32}:Tb^{3+}$-containing one, and even higher than the pristine hydrogel. Comparatively, an $HNTs@Na_5Lu_9F_{32}:Tb^{3+}$-containing hydrogel should have a better application potential in X-ray scintillator screens than the $OA@Na_5Lu_9F_{32}:Tb^{3+}$-based analogues.

Supplementary Fig. 15 displays a series of photographs of the $HNTs@Na_5Lu_9F_{32}:Tb^{3+}$-containing hydrogel under different tensile strains and bending angles. It can be bent to 90° without any breaking and the elongation can be increased from 0% to 250% without rupture. Moreover, the feasibility of X-ray imaging by the $HNTs@Na_5Lu_9F_{32}:Tb^{3+}$-containing hydrogel is also demonstrated based on the capsule model. Anchoring nanoscintillators onto the surface of HNTs may pave an alternative path in developing flexible scintillator materials with good mechanical properties.

## Information encryption in multilayer hydrogels

Hydrogels that can changing their color in response to external stimuli such as UV, heat, and pH have been proposed for storing of information[50,51]. However, information decryption under single stimulation is easy to be decoded, stolen, and faked[52], which inspires us to explore multilevel encryption hydrogels based on the different emission behaviors between X-ray-excited RL and photoluminescence (PL) (Supplementary Table 4). Some organic materials composed of light atoms exhibit weak X-ray absorption, which can only generate luminescence under UV light and cannot emit light under X-ray irradiation[1,23,53]. Such differences in emission behavior inspired us to combine an anti-counterfeiting technology and multilevel encryption into the hybrid hydrogel with encrypted information.

To implement this, three kinds of hydrogels are synthesized and the schematic diagram is shown in Fig. 8. **Gel-0** is prepared by using 1,4-phenyldiboronic as the crosslinker for polyvinyl alcohol (PVA) and is used as the substrate to prepare multilayer hydrogels, which cannot emit any luminescence under UV or X-ray. $HNTs@Na_5Lu_9F_{32}:Tb^{3+}$-containing hydrogels (**Gel-1**) are prepared in letter-type molds. The boronic acid functionalized tetraphenylethylene (**M3**) is employed, which gives rise to the three-dimensional network in **Gel-1**. The presence of both

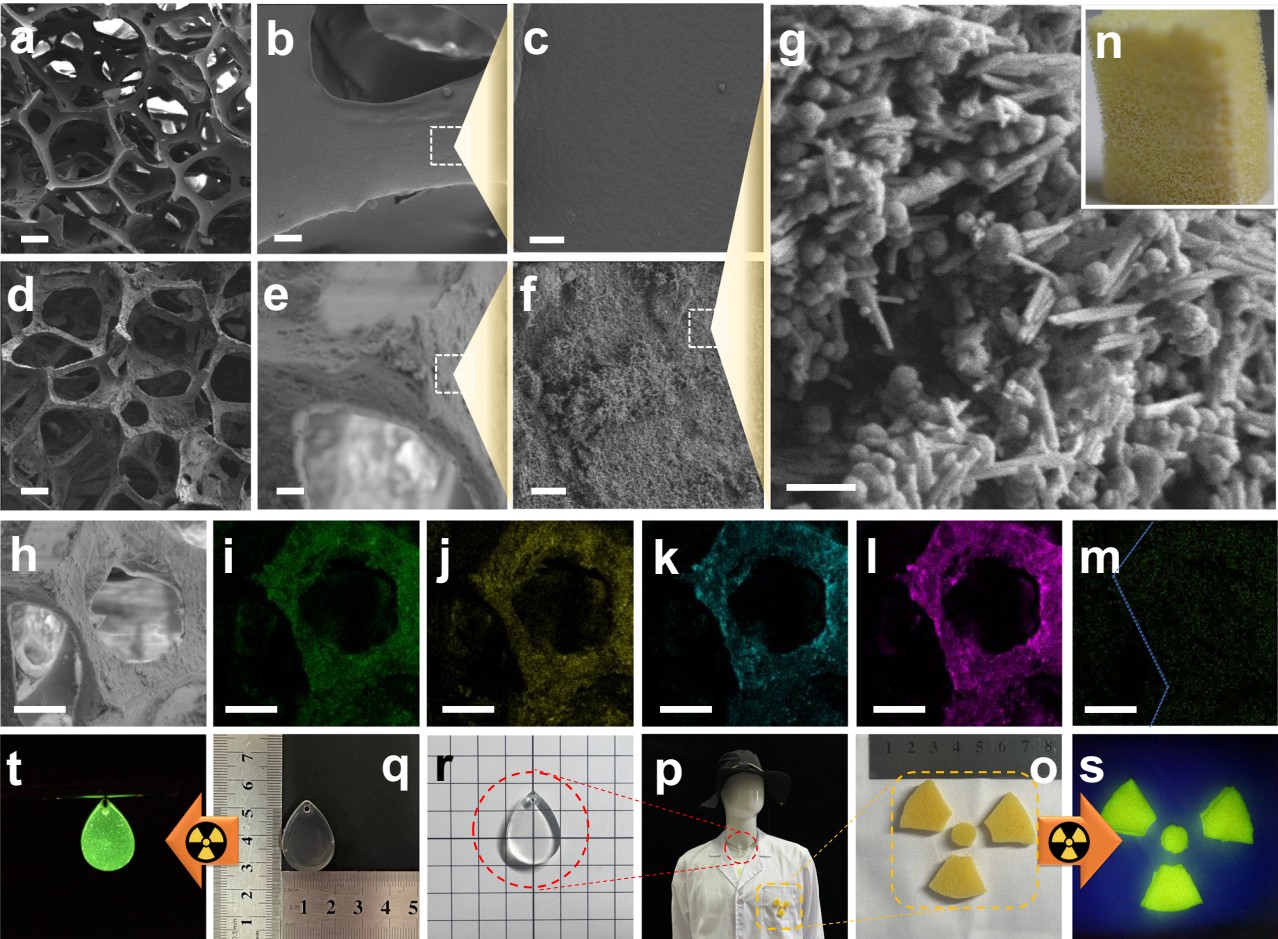

**Fig. 5 | Micromorphology results and application in monitoring X-ray.**
**a** Scanning electron microscope (SEM) images of pristine polyurethane foam (PUF) (bar: 200 μm). **b** SEM images of pristine PUF (bar: 20 μm). **c** SEM images of pristine PUF (bar: 5 μm). **d** SEM images of $Tb^{3+}$-doped $Na_5Lu_9F_{32}$:$Tb^{3+}$ anchored halloysite nanotubes coated polyurethane foam (HNTs@$Na_5Lu_9F_{32}$:$Tb^{3+}$@PUF) (bar: 200 μm). **e** SEM images of HNTs@$Na_5Lu_9F_{32}$:$Tb^{3+}$@PUF (bar: 20 μm). **f** SEM images of HNTs@$Na_5Lu_9F_{32}$:$Tb^{3+}$@PUF (bar: 5 μm). **g** SEM images of HNTs@$Na_5Lu_9F_{32}$:$Tb^{3+}$@PUF (bar: 500 nm). **h** SEM images of HNTs@$Na_5Lu_9F_{32}$:$Tb^{3+}$@PUF (bar: 100 μm). **i–m** Elemental mapping images of HNTs@$Na_5Lu_9F_{32}$:$Tb^{3+}$@PUF (**i** Al, **m** Si, **k** Na, **l** F, and **m** Tb; bar: 100 μm). **n** Photograph of HNTs@$Na_5Lu_9F_{32}$:$Tb^{3+}$@PUF taken under normal light. **o** Photograph of HNTs@$Na_5Lu_9F_{32}$:$Tb^{3+}$@PUF fabricated into desired shapes. **p** Photograph of the mannequin wearing special-made lab coat and drop shape pendant. **q** Photograph of the drop shape pendant under normal light. **r** Illustration of the transparency of the drop shape pendant. **s** Photograph of HNTs@$Na_5Lu_9F_{32}$:$Tb^{3+}$@PUF taken under X-ray (3.1 cGy s$^{-1}$). **t** Photograph of the drop shape pendant taken under X-ray (3.1 cGy s$^{-1}$).

tetraphenylethylene (TPE) units and HNTs@$Na_5Lu_9F_{32}$:$Tb^{3+}$ makes **Gel-1** exhibit a strong blue luminescence due to the aggregation-induced emission (AIE) effect[54,55], and also endows RL behavior to **Gel-1**. **Gel-2** is prepared in a similar manner without the addition of HNTs@$Na_5Lu_9F_{32}$:$Tb^{3+}$ making **Gel-2** only photoluminescent. **Gel-1** and **Gel-2** are prepared in letter shapes to compose the information layer. The presence of boronic ester crosslinks and potential residual arylboronic acid in the hydrogels induce self-healing properties and dynamic exchange behavior, leading to merging of the individual hydrogel layers. Herein, the design of a sandwich-like multilayer hydrogel with encrypted information is proposed (Fig. 9a), which consists of two outer non-luminescence layers (**Gel-0**) and an inner information layer (sequence: **Gel-0**/ information layer /**Gel-0**).

The obtained multilayer hydrogel (Fig. 9b) can be folded or bent into nearly a U-shape without any detachment of the hydrogel layers indicating efficient crosslinking between the layers (Fig. 9c, d). Rheological investigations reveal that the storage modulus exceeded the loss modulus an angular frequency ranging from 0 to 100 rad s$^{-1}$ (Supplementary Fig. 17), also suggesting efficient dynamic exchange of boronic ester crosslinks between the hydrogel layers.

The obtained multilayer hydrogel cannot show any information under normal light conditions (Fig. 9b). When exposed to UV light, the letters EHBUT illuminate and can be clearly observed in Fig. 9e. No differences can be found among the luminescence from the five letters. The information recognized from UV light is defined as false information. The encrypted letters (HBU shown in Fig. 9f) can only be identified with exposure to X-ray.

To the best of our knowledge, there is no report on hydrogels for information encryption by using X-ray as the decoding tool. Besides this, the multilayer hydrogels support an extra safety encryption technology to prevent information leakage and combat fakes, because the false information can be read from multilayer hydrogels under UV light that serves as a commonly-used decoding tool in conventional anti-counterfeiting and encryption technologies. The multilayer hydrogels are assembled based on the dynamic covalent bonds rather than physically attached, which can afford uniform hydrogels and further allows more complicated programming for information camouflage and multilevel encryption based on the three different types component hydrogels: non-emission (**Gel-0**), only PL (**Gel-2**), and PL&RL (**Gel-1**). The presence of dynamic covalent bonds in the

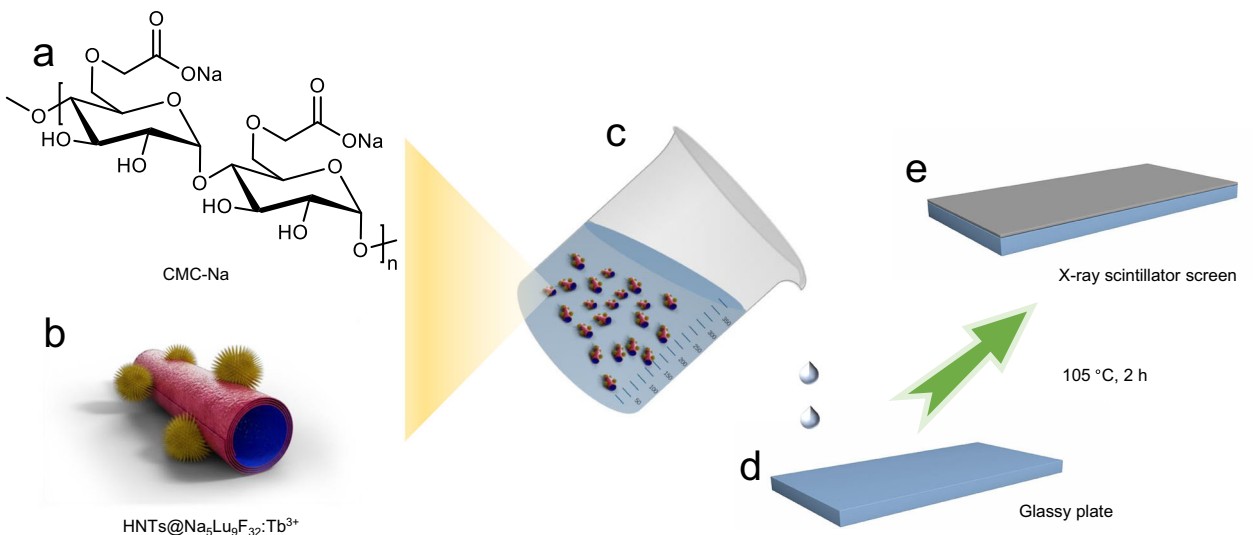

**Fig. 6 | Preparation of rigid X-ray scintillator screen. a** Chemical structure of carboxymethylcellulose sodium (CMC-Na); **b** $Tb^{3+}$-doped $Na_5Lu_9F_{32}$ anchored halloysite nanotubes (HNTs@$Na_5Lu_9F_{32}$:$Tb^{3+}$); **c** HNTs@$Na_5Lu_9F_{32}$:$Tb^{3+}$-contained CMC-Na solution; **d** Glass plate (7.6 cm × 2.5 cm); **e** X-ray scintillator screen obtained by dropping HNTs@$Na_5Lu_9F_{32}$:$Tb^{3+}$-contained CMC-Na solution onto the glass plate and then dried at 105 °C.

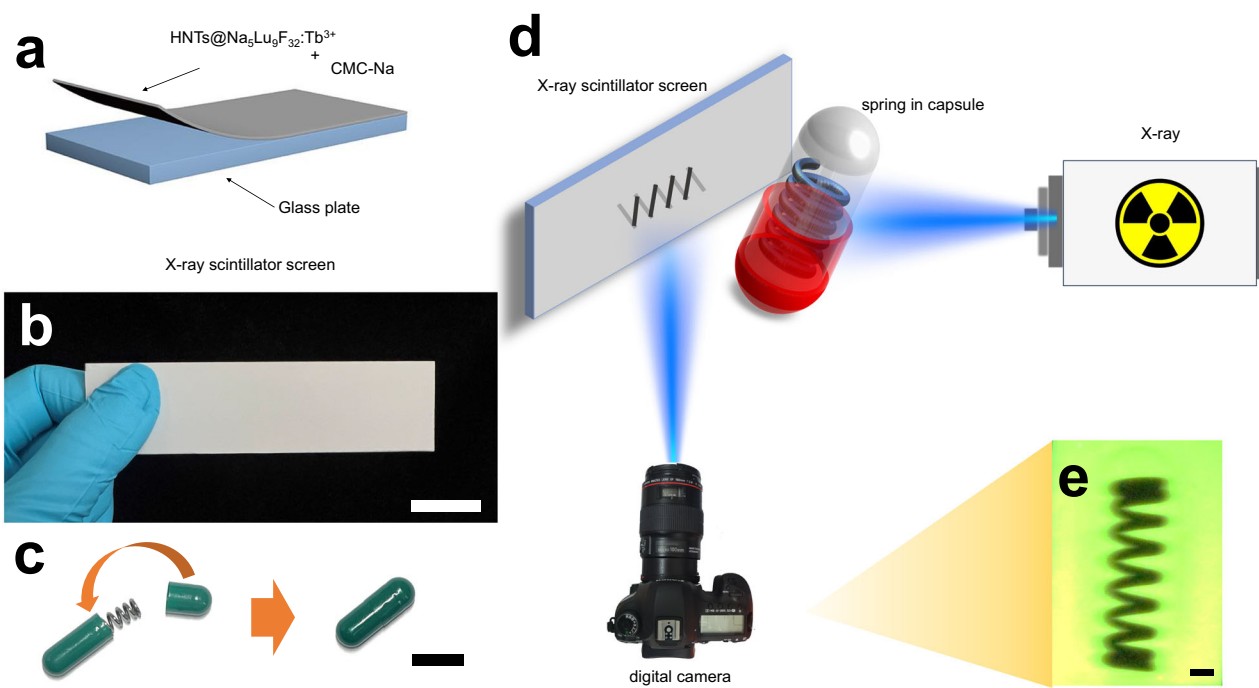

**Fig. 7 | X-ray imaging studies. a** Scheme of the production and composition of the X-ray scintillator screen (CMC-Na is abbreviated from carboxymethylcellulose sodium; HNTs@$Na_5Lu_9F_{32}$:$Tb^{3+}$ is abbreviated from $Tb^{3+}$-doped $Na_5Lu_9F_{32}$ anchored halloysite nanotubes). **b** Photograph of the as-prepared X-ray scintillator screen (Thickness: ca.1 mm; bar: 2 cm). **c** Fabrication of the spring-containing capsule model (bar: 1 cm). **d** Schematic diagram of the X-ray imaging system based on the as-prepared X-ray scintillator screen. **e** The image relating to the capsule model taken from the digital camera (X-ray: 30 kV, 20 mA; Power: 600 W; bar: 2 mm).

obtained multilayer hydrogels would also contribute to potential self-healing properties.

In summary, $Na_5Lu_9F_{32}$:$Tb^{3+}$ is successfully generated on the surface of HNTs-CA to afford HNTs@$Na_5Lu_9F_{32}$:$Tb^{3+}$ an X-ray scintillator with good water-dispersibility, high sensitivity to X-ray irradiation, low toxicity, and desirable compatibility with various polymer matrixes. The HNTs@$Na_5Lu_9F_{32}$:$Tb^{3+}$ exhibits a typical X-ray-responsive scintillator behavior and displays a strong emission peak at 544 nm upon exposure to X-ray irradiation, enabling on-off switching of the RL by on-off switching of the X-ray irradiation. The application potential and aqueous processability of the obtained HNTs@$Na_5Lu_9F_{32}$:$Tb^{3+}$ is demonstrated by water-based fabrication into an X-ray-sensitive polyurethane foam, X-ray-sensitive transparent composite, rigid and flexible X-ray imaging screens, and multilayer hydrogels for information encryption, which can only be readout by X-ray irradiation.

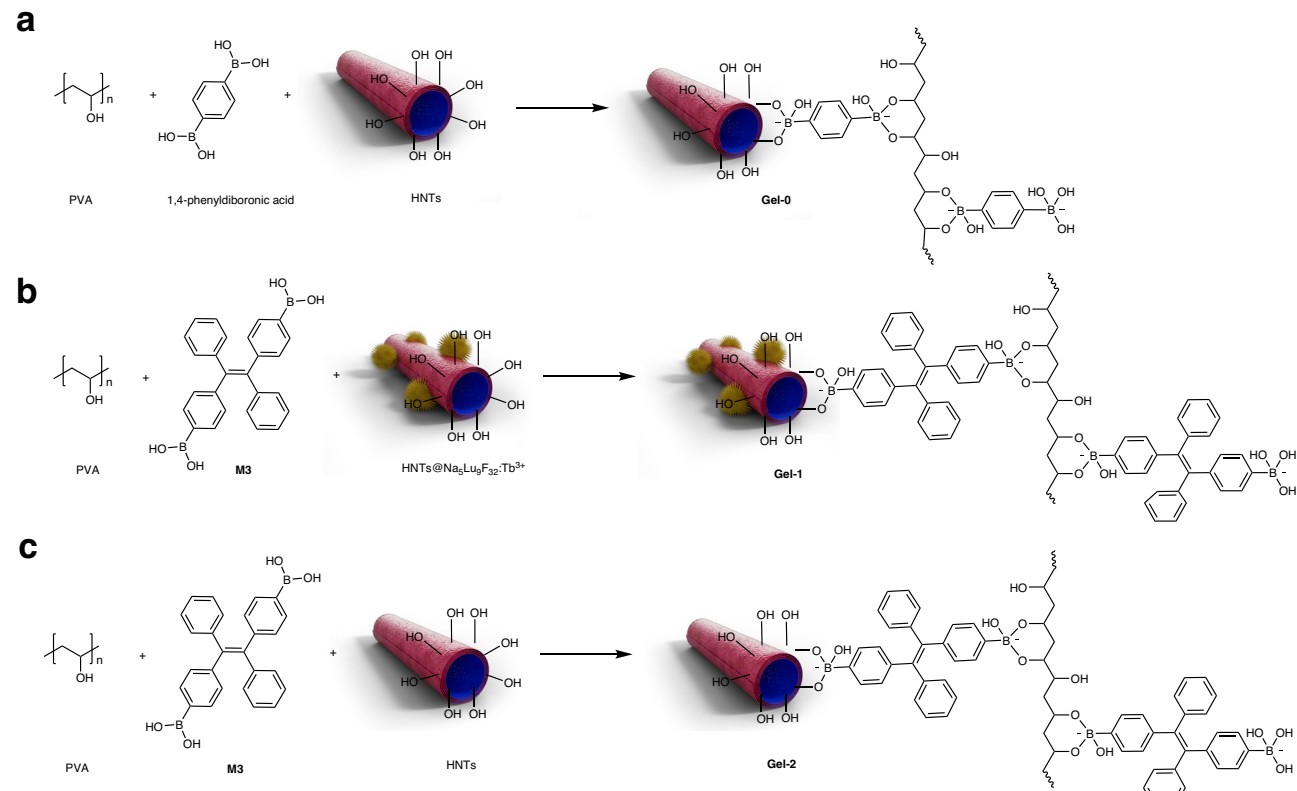

**Fig. 8 | Synthesis of hydrogels in information encryption study. a Gel-0** is prepared in cuboid and exhibits no emission (PVA is abbreviated from polyvinyl alcohol; HNTs is abbreviated from halloysite nanotubes). **b Gel-1** is prepared in letter sharp (HBU), which can emit blue luminescence under UV light (365 nm) and green luminescence under X-ray, respectively (HNTs@Na$_5$Lu$_9$F$_{32}$:Tb$^{3+}$ is abbreviated from Tb$^{3+}$-doped Na$_5$Lu$_9$F$_{32}$ anchored halloysite nanotubes; **M3** represents boronic acid functionalized tetraphenylethylene). **c Gel-2** is prepared in letter sharp (ET) and emit blue luminescence under UV light (365 nm).

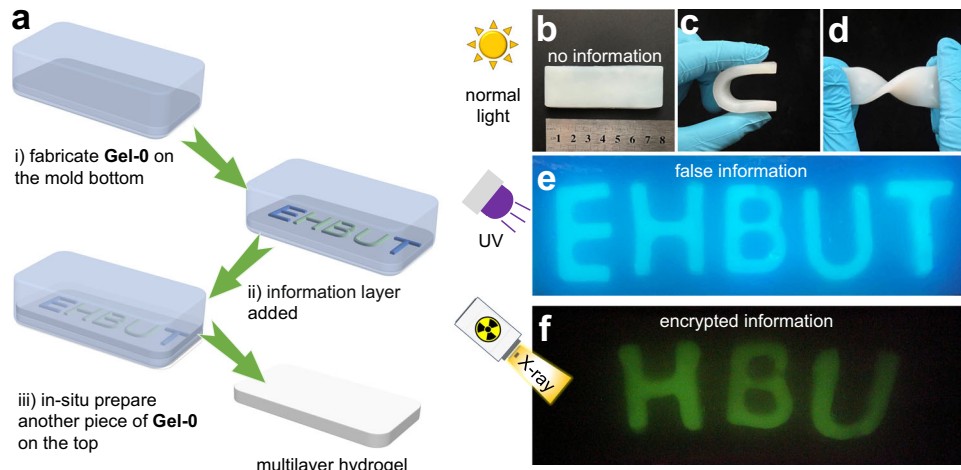

**Fig. 9 | Information encryption studies. a** Scheme of preparation of multilayer hydrogels. **b** Normal version of the prepared multilayer hydrogel. **c** Bended hydrogel sample. **d** Twisted hydrogel sample. **e** Photograph of the false information EHBUT in the prepared multilayer hydrogel under UV light (365 nm). **f** Photograph of the encrypted information HBU in the multilayer hydrogel after exposure to X-ray (30 kV, 20 mA).

Altogether it is demonstrated that the developed HNTs@Na$_5$Lu$_9$F$_{32}$:Tb$^{3+}$ nanocrystals are efficient X-ray scintillators that are easily processable as aqueous dispersion to develop X-ray sensitive flexible substrates, coatings, and hydrogels. Hence the reported work provides insights for X-ray-sensitive soft materials, while also demonstrates the application potential of clay-based nanomaterials as support for functional nanoparticles.

## Methods

The description of materials and characterization is included in the Supplementary Note 1 and Supplementary Note 2, respectively.

### Synthesis of citric acid-modified halloysite nanotubes

5.0 g HNTs-NH$_2$ was added into the DMSO solution containing citric acid (1.92 g, 10 mmol), 1-(3-dimethylaminopropyl)-3-ethylcarbodiimide

hydrochloride (6.2 g, 34.5 mmol), 1-hydroxybenzotriazole (4.05 g, 30 mmol), and N,N-diisopropylethylamine (3.9 g, 30 mmol). Then the solution was stirred at room temperature for 48 h. The residue was collected by centrifugation and then washed with 5% HCl solution, saturated $Na_2CO_3$ solution, saturated NaCl solution, ethanol, and acetone in sequence. The obtained solid was dried in vacuum to give HNTs-CA as a white solid.

## Synthesis of water-dispersible X-ray scintillator
1.50 g HNTs-CA was accurately weighed and added to an aqueous solution containing $NaNO_3$ (0.51 g, 6.0 mmol), $Lu(NO_3)_3 \cdot 6H_2O$ (4.69 g, 10.0 mmol), and $Tb(NO_3)_3 \cdot 6H_2O$ (0.45 g, 1.0 mmol). The system was stirred at a low speed of 200 rpm for 30 min to achieve a homogeneous suspension. 1.48 g $NH_4F$ was added into the mixture. After being regulated to weak acid with a pH value of 5.0 by nitric acid, the suspension was stirred for another 30 min and then transferred into a hydrothermal reactor with a filling rate of 70%. The hydrothermal reactor was heated at 180 °C for 24 h. After cooling to room temperature, the suspension was picked out and subjected to centrifugation. The collected residue was thoroughly washed with water and ethanol followed by being dried in vacuum to afford $HNTs@Na_5Lu_9F_{32}:Tb^{3+}$ as a white solid.

## Preparation of X-ray-sensitive polyurethane foam
PUF pieces (3.5 cm × 2.5 cm × 1.5 cm) were rinsed thoroughly with water and then dried under reduced pressure overnight. $HNTs@Na_5Lu_9F_{32}:Tb^{3+}$ was added into water to achieve $HNTs@Na_5Lu_9F_{32}:Tb^{3+}$ aqueous suspension (ca. 100 mg mL$^{-1}$). The PUF pieces were immersed in the suspension, thoroughly compressed three times, and soaked for 3 min. Then the foam was picked out from the suspension and wrung out by using mechanical rollers. After drying in vacuum, the composite foam $HNTs@Na_5Lu_9F_{32}:Tb^{3+}@PUF$ was obtained and stored in a dry box before use.

## Preparation of X-ray-sensitive epoxy resin composite
20 mg $HNTs@Na_5Lu_9F_{32}:Tb^{3+}$ were dispersed in 10 mL of anhydrous ethanol and then added into the E51 solution (2 g in anhydrous ethanol). The mixture was sonicated for 5 min and then stirred at 70 °C for 2 h to obtain a well-dispersed suspension. The anhydrous ethanol was removed by rotary evaporation. The sample was weighted and the polyamide curing agent was added to the solution with a weight ratio of 1:1. The system was mixed by thorough agitation and then poured into the drop-shaped mold. The liquid-filled mold was degassed in a vacuum oven for 10 min and then placed at room temperature for 24 h to afford the $HNTs@Na_5Lu_9F_{32}:Tb^{3+}@epoxy$ resin composite.

## Preparation of X-ray scintillator screens
The $HNTs@Na_5Lu_9F_{32}:Tb^{3+}$ powders were filtered by a 200-mesh sieve. 2.0 g filtered powders were added into a 0.5 wt% CMC-Na solution (20 mL). Vigorous stirring was continued to afford a uniform suspension. This suspension was poured onto a glass plate (7.6 cm × 2.5 cm) and then spread uniformly via a mild shock by using a shaker rotating with the agitation rate at 100 rpm. The samples were placed in an oven at 105 °C for 2 h and then carefully polished. The obtained screens were stored in a dry box before used.

The flexible screen is prepared following another method. PVA solution was prepared by adding 1.35 g PVA in 12.0 mL water. Intense stirring is conducted at 80°C until it is completely dissolved. 65 mg $HNTs@Na_5Lu_9F_{32}:Tb^{3+}$ was then added and the stirring is continued to afford a uniform suspension. The mixture was poured into the mold with the addition of 1,4-phenyldiboronic acid-containing NaOH solution ($4.8 \times 10^{-3}$ mmol mL$^{-1}$) under argon and then heated at 80 °C for 1 h. After cooling to room temperature, the gel was picked out from the mold and cut into demanded sizes for imaging studies.

## Preparation of multilayer hydrogels
PVA solution was prepared according to the above-mentioned method. 130 mg $HNTs@Na_5Lu_9F_{32}:Tb^{3+}$ was then added and intense stirring was employed to afford a uniform suspension. 1.5 mL of a **M3**-containing NaOH solution ($4.8 \times 10^{-3}$ mmol mL$^{-1}$, pH = 10.5) was poured into the $HNTs@Na_5Lu_9F_{32}:Tb^{3+}$-containing suspension with stirring for 5 min. The mixture was transferred into tailor-made molds and then frozen at −20 °C for 2 h. After defrosting at room temperature for 1 h, the self-standing hydrogel (**Gel-1**) can be obtained with letter shapes (H, B, and U). **Gel-2** was also prepared in letter-like molds following a similar procedure with the replacement of $HNTs@Na_5Lu_9F_{32}:Tb^{3+}$ to pristine HNTs to give the hydrogels with shape for the letter E and T.

**Gel-0** was prepared in a rectangular mold (7.5 cm × 2.5 cm) by using PVA and HNTs as substrate material. 1,4-phenyldiboronic acid was used as the crosslinking agent. To prepare the multilayer hydrogel, the **Gel-0** was placed on the bottom of the rectangular mold. Then **Gel-1** and **Gel-2** samples were added onto **Gel-0** piece following the sequence: E-H-B-U-T. Another layer of **Gel-0** was prepared in situ on the top of the layer containing **Gel-1** and **Gel-2** to give the hydrogel multilayer system. Then the overlaid samples were heated at 80 °C for 30 min to give the sandwich-like multilayer hydrogels.

## Reporting summary
Further information on research design is available in the Nature Portfolio Reporting Summary linked to this article.

## Data availability
The data generated in this study are provided in Supplementary Information/Source Data file.

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

## Acknowledgements

H.Z. acknowledges the support from National Natural Science Foundation of China (Grant No. 22102045) and the Chinese Scholarship Council (No. 202108130104). R.H. acknowledges continuous financial support from Ghent University and the Research Foundation - Flanders (FWO). Y. Y. acknowledges the support from National Natural Science Foundation of China (Grant No. 12374373).

## Author contributions

R.H., H.Z., and Y.Y. conceived and supervised the project. H.Z., X.B., and Y.W. designed the synthesis. B.Z. synthesized the materials. Y.Y. and C.C. performed radioluminescence measurement. Yu Wang and Yuan Wang performed the additional work on synthesis and characterizations during the revision stage. R.H. and H.Z. wrote the manuscript. H.Z., B.Z., and K.Z. participated in data analysis. All authors participated in the discussion and analysis of the manuscript.

## Competing interests

The authors declare no competing interests.
