## [Peer Review File · Nature Communications]

Water-dispersible X-ray scintillators enabling coating and blending with polymer materials for multiple applicationsREVIEWER COMMENTS

Reviewer #1 (Remarks to the Author):

In this manuscript, Zhang et al. synthesized halloysite nanotubes modified with citric acid (HNTs@Na₅Lu₉F₃₂:Tb³⁺), which exhibit excellent water dispersion and high sensitivity to X-ray luminescence. Overall, this study demonstrates the potential application of clay-based nanomaterials as functional nanoparticle carriers, and the paper has a certain integrity. However, this study lacks novelty and significant breakthroughs in this research area, as numerous methods for surface modification of nanoscintillators have already been reported for biomedical applications. Furthermore, the incorporation of nanocrystal scintillators into polymers for X-ray imaging and information encryption has been previously demonstrated. As a result, I cannot recommend publication in a prestigious journal such as Nature Communications. Please refer to the detailed comments below for your consideration:

1. In the results and discussion section, the authors dedicate a significant portion of the discussion to the synthesis and characterization of HNTs-CA and HNTs@Na₅Lu₉F₃₂:Tb³⁺, addressing the question of whether their proper removal can be achieved and emphasizing its importance.
2. Additionally, in scheme 2, the PUF sample was immersed in an aqueous dispersion of the HNTs@Na₅Lu₉F₃₂:Tb³⁺ X-ray scintillator, followed by the removal of excess water through thorough compression. However, it is noted in the article (Appl Phys, A Mater Sci Process. 2007;88(2):391–5.) that a well-dispersed HNT tends to reaggregate in the polymer during the drying process. The article mentions that three repetitions can achieve a uniform coating, but it is unclear how this uniformity is precisely defined. Furthermore, the PUF load should vary for each soaking process.
3. In lines 279-285, the author discusses the potential application of HNTs@Na₅Lu₉F₃₂:Tb³⁺@PUF material, primarily focusing on its use in the production of protective clothing or X-ray protective walls. While this application appears feasible to some extent, the paper does not explore the material's utilization beyond this point. Instead, it merely provides a brief description of possible future application fields. This aspect could have been presented as a separate chapter in the paper, as its current treatment lacks sufficient evidence and persuasive argumentation.
4. Flexible scintillator screens have the advantage of enabling high-quality imaging on non-planar objects and offering a broader range of applications compared to rigid screens (Adv. Mater. 2022, 34, 2204801.). In this study, a rigid scintillator screen was prepared on a glass plate using a similar method as TLC plate preparation. It raises the question of whether the same procedure can be employed to prepare flexible polymer-based scintillator screens, allowing for mass production. Additionally, it is worth noting that the image quality achieved with scintillator screens in this work is considered mediocre.
5. In Section 2.4, the author uses HNTs@Na₅Lu₉F₃₂:Tb³⁺ to make multi-layer hydrogel for information encryption. In fact, the use of hydrogels for information encryption has been extensively studied, but I'm not sure what new benefits and breakthroughs this protocol provides.

Reviewer #3 (Remarks to the Author):

The manuscript presents the synthesis of Na₅Lu₉F₃₂:Tb³⁺ nanoparticles (NPs) on surface modified-halloysite nanotubes, which are applied as X-ray scintillators in radioluminescent polymer materials, X-ray imaging coatings and information encrypting hydrogels.

The work presents novel results. It should be noted that Na₅Lu₉F₃₂:Tb³⁺, while a known material, has not been previously reported as X-ray scintillator. Moreover, the deposition of X-ray scintillators on halloysite nanotubes is also here reported for the first time (even though composites of X-ray scintillators with SiO₂ have already been reported). Another noteworthy approach is the preparation of a multilayer hydrogels containing the synthesized composite material, which can be used for information encryption.

The significance of these approaches is more dubious. One of the main aims of the work, as described by the authors in the Introduction, is the development of “water-dispersible X-ray scintillators”. The solution here adopted is based on the deposition of the X-ray scintillator nanoparticles on a surface-modified clay material (citrate-APTES-grafted halloysite nanotubes, HNT). It is unclear, however, why the authors did not simply modify via covalent grafting with the same organic moieties the surface of the Na₅Lu₉F₃₂:Tb³⁺ nanoparticles. Furthermore, the use of halloysite nanotubes is not properly justified. The authors claim that halloysite nanotubes are non-toxic, however contradicting literature studies have been reported (see e.g., *Toxicological Research* 37, 301–310 (2021)). The authors do not exploit the characteristic structural and morphological features of HNTs, since they simply deposit the NPs on the external surface of the nanotubes. Silica NPs or other clay minerals could have been used instead, especially considering that those materials are already widely used commercially as polymer fillers and in surface coatings.

The authors present extensive characterization data, particularly concerning the structural and morphological properties of the HNTs@Na₅Lu₉F₃₂:Tb³⁺. However, the authors claimed that “Na₅Lu₉F₃₂:Tb³⁺ is introduced as new X-ray scintillating material” but the characterization of the main performance parameters of this new scintillator lacks important data (e.g., conversion efficiency, decay time, PL emission). Furthermore, the stability of the composite material with polyurethane foam and of the X-ray scintillator screen should be investigated in a more quantitative fashion.

Data discussion could be improved. The image of a single functionalized nanotube is reported: this is not sufficient to prove the homogeneity of the sample. No estimate about the average size of the NPs is reported. Discussion of several characterization techniques (XPS, XRD, TGA...) is only qualitative. Furthermore, some details should be revised. For instance, in solid state NMR spectra, it makes no sense to report signal position with two decimal digits. In Figure S7, please double check the scale bars (the panel on the right seems to report a larger magnification, but the scale bar does not support that).

Overall, the manuscript could possibly be published pending major revisions.

Reviewer #4 (Remarks to the Author):

This paper reports on a new nanoparticle X-ray scintillator: HNTs@Na₅Lu₉F₃₂:Tb³⁺ prepared by hydrothermal synthesis. The radioluminescent Na₅Lu₉F₃₂:Tb³⁺ nanoparticles were decorated on HNTs nanotubes and incorporated in three different composite platforms for potential scintillation applications. A high light yield of 15800 photons MeV⁻¹ is reported for Na₅Lu₉F₃₂:Tb³⁺ nanoparticles. The high water dispersibility of the particles facilitates its processing in organic or inorganic matrices for large scale manufacturing. The new material seems promising and its versatile processing capability is interesting for X-ray

sensing or imaging applications.

It is not clear how the X-ray attenuation coefficient for the scintillator and LuAG:Ce scintillator are determined for the calculation of the light yield. Is Na₅Lu₉F₃₂:Tb³⁺ material only considered for attenuation coefficient calculation or the whole screen scintillator? Information on sample thickness and volume, density, X-ray energy etc. should be provided.

A more detailed characterization on the structure and properties of Na₅Lu₉F₃₂:Tb³⁺ nanoparticle scintillator should be presented. For example, the crystalline structure, density, decay time etc. And a comparison with other nanoparticle scintillators (perovskites, NaLuF₄:Tb³⁺ etc) reported should be included. Does any persistent radioluminescence as in NaLuF₄:Tb³⁺ observed from this new structure?

What is the X-ray energy used for X-ray imaging studies in Figure 4? What's the thickness of screen?

Define "low-dose X-ray" that is used several times in the manuscript. What is the X-ray energy and what is the dosage.

What is the average size of Na₅Lu₉F₃₂:Tb³⁺ nanoparticles and HNTs@Na₅Lu₉F₃₂:Tb³⁺. One advantage of nanoparticle scintillator is low scattering and the potential for the formation of transparent composite scintillator. Could the authors comment on the transparency of screen or other composites structures prepared to maximize light output?

Reviewer #1 (Remarks to the Author):

In this manuscript, Zhang et al. synthesized halloysite nanotubes modified with citric acid (HNTs@Na₅Lu₉F₃₂:Tb³⁺), which exhibit excellent water dispersion and high sensitivity to X-ray luminescence. Overall, this study demonstrates the potential application of clay-based nanomaterials as functional nanoparticle carriers, and the paper has a certain integrity. However, this study lacks novelty and significant breakthroughs in this research area, as numerous methods for surface modification of nanoscintillators have already been reported for biomedical applications. Furthermore, the incorporation of nanocrystal scintillators into polymers for X-ray imaging and information encryption has been previously demonstrated. As a result, I cannot recommend publication in a prestigious journal such as Nature Communications. Please refer to the detailed comments below for your consideration:

Response: Thanks for your valuable comments. We have performed a series of additional experiments and carefully revised the manuscript to more clearly illustrate the novelty and breakthroughs of the reported work. There are indeed many surface modification methods in the literature that focus on improving the water-dispersibility and biocompatibility of the nanoscintillators. In these studies, oleic acid (OA), polyethylene glycol (PEG), and folic acid are usually employed to achieve the surface modification of nanoscintillators. We also made attempts to modify the surface of Na₅Lu₉F₃₂:Tb³⁺ with oleic acid to give the product oleic acid-modified Na₅Lu₉F₃₂:Tb³⁺ (OA@Na₅Lu₉F₃₂:Tb³⁺). The obtained OA@Na₅Lu₉F₃₂:Tb³⁺ was confirmed by XPS and the water-dispersibility could be improved to some extent. However, after being incorporated into the hydrogel, it gives rise to a lowered tensile property for the obtained hydrogel. Anchoring Na₅Lu₉F₃₂:Tb³⁺ on the surface of halloysite nanotubes (HNTs) can effectively solve the problem. The HNTs@Na₅Lu₉F₃₂:Tb³⁺-containing hydrogel shows a much better tensile strength than the OA@Na₅Lu₉F₃₂:Tb³⁺-containing hydrogel, even higher than the pristine hydrogel, which is likely related to orientation of the HNTs during the stretching process, which is known to improve the tensile strength (*Prog. Polym. Sci.* 38, 1690-1719 (2013)). Tensile strength is quite important in developing flexible scintillator screens. Therefore, anchoring nanoscintillators onto the surface of HNTs may pave an alternative path in developing flexible scintillator materials with good mechanical properties.

We also redesigned the experiments on information encryption hydrogels to make the novelty more clear by introducing the concept of “false information” under UV light. Up to now, the majority of information encryption studies on hydrogels have focused on using UV light as a decryption tool. It should be noted that some materials can exhibit different emission behavior between X-ray and UV light. For some organic materials, it can only generate luminescence under UV light while cannot emit light under X-ray. It inspired us to develop X-ray-decrypted hydrogels consisting of different kinds of light-emitting materials. In newly designed study, the obtained multilayer hydrogel can give “false information” under UV light, in which the “encrypted information” can only be recognized by X-ray. By utilizing the different emission

behaviors under X-ray and UV light, the anti-counterfeiting technology is introduced in the hydrogels for information encryption. Following this way, X-ray-decrypted hydrogels can afford a securer way to protect the information.

Hence based on this further discussion and additional experiments, we respectfully disagree with the reviewer that the work is not suitable for publication in Nature Communications. Importantly, reviewers #3 and #4 also supported publication of our work in Nature Communications after revision.

Please see the detailed point-by-point responses in the following.

1. In the results and discussion section, the authors dedicate a significant portion of the discussion to the synthesis and characterization of HNTs-CA and HNTs@Na₅Lu₉F₃₂:Tb³⁺, addressing the question of whether their proper removal can be achieved and emphasizing its importance.

Response: Thanks for your comment. We realized that we indeed used a lengthy description of the synthesis and characterizations. The discussion relating to the synthesis and characterization of HNTs-CA has been moved to Sec. S2.1 in the revised supplementary information to simplify and streamline the main text.

2. Additionally, in scheme 2, the PUF sample was immersed in an aqueous dispersion of the HNTs@Na₅Lu₉F₃₂:Tb³⁺ X-ray scintillator, followed by the removal of excess water through thorough compression. However, it is noted in the article (Appl Phys, A Mater Sci Process. 2007;88(2):391–5.) that a well-dispersed HNT tends to reaggregate in the polymer during the drying process. The article mentions that three repetitions can achieve a uniform coating, but it is unclear how this uniformity is precisely defined. Furthermore, the PUF load should vary for each soaking process.

Response: We thank the reviewer for this comment and have carefully studied the cited article (Appl Phys, A Mater Sci Process. 2007;88(2):391–5) to be able to assess whether reaggregation occurs in our system. In this article, the reaggregation behavior occurs in a polyvinyl alcohol (PVA)/HNTs solution. PVA can be dissolved in water at 70 °C as shown in the following Fig. R1A&B. When cooling to room temperature, the PVA/HNTs solution would undergo a Sol-Gel (Fig. R1C) transformation. After then, the evaporation of water may result in the shrinking behavior of PVA chains, which further cause the reaggregation of HNTs. However, in our study, the PUF samples cannot be dissolved in water solution. There is no Sol-Gel transformation process or chain shrinking behaviors from 70 to 25 °C (Fig. R1D&F). The SEM observations show the HNTs@Na₅Lu₉F₃₂:Tb³⁺@PUF exhibits similar pore size as compared to that of pristine PUF (Fig. 3A&D). Moreover, no serious aggregation behavior was observed. We also measured the RL emission spectrum of HNTs@Na₅Lu₉F₃₂:Tb³⁺@PUF (Fig. S13A), which shows a similar tendency to those of HNTs@Na₅Lu₉F₃₂:Tb³⁺ and Na₅Lu₉F₃₂:Tb³⁺. All of the characteristic peaks of HNTs@Na₅Lu₉F₃₂:Tb³⁺ can be

clearly detected in the RL emission spectrum of HNTs@Na₅Lu₉F₃₂:Tb³⁺@PUF. No newly appeared peaks were detected.

To further demonstrate the uniformity of the coating behaviors, five sites in the obtained HNTs@Na₅Lu₉F₃₂:Tb³⁺@PUF sample were randomly selected and the peak intensity of 544 nm in the radioluminescence spectra of each site was recorded. The relative standard deviation (RSD) value of the results from the intensity values is calculated as 4.8%, suggesting a uniform coating of HNTs@Na₅Lu₉F₃₂:Tb³⁺ on the PUF surface. The results and discussion have been added in Line 259-263 in the revised manuscript.

So, we demonstrate here that the preparation process of HNTs@Na₅Lu₉F₃₂:Tb³⁺@PUF does not cause aggregation of the HNTs and is substantially different from the reaggregation behavior reported in Appl Phys, A Mater Sci Process. 2007;88(2):391–5.

Fig. R1. Illustration of the difference between PVA and PUF in HNTs@Na₅Lu₉F₃₂:Tb³⁺ suspension. A PVA in HNTs@Na₅Lu₉F₃₂:Tb³⁺ suspension at 70°C. B PVA in water at 70°C. C PVA in HNTs@Na₅Lu₉F₃₂:Tb³⁺ suspension at 25°C. D PUF in HNTs@Na₅Lu₉F₃₂:Tb³⁺ suspension at 70°C. E PUF in HNTs@Na₅Lu₉F₃₂:Tb³⁺ suspension at 25°C.

Fig. S13A. X-ray-excited RL emission spectra of the HNTs@Na₅Lu₉F₃₂:Tb³⁺@PUF samples before and after stability test.

3. In lines 279-285, the author discusses the potential application of

HNTs@Na₅Lu₉F₃₂:Tb³⁺@PUF material, primarily focusing on its use in the production of protective clothing or X-ray protective walls. While this application appears feasible to some extent, the paper does not explore the material's utilization beyond this point. Instead, it merely provides a brief description of possible future application fields. This aspect could have been presented as a separate chapter in the paper, as its current treatment lacks sufficient evidence and persuasive argumentation.

Response: We thank the reviewer for this nice suggestion. We have performed additional experiments to fabricate the obtained HNTs@Na₅Lu₉F₃₂:Tb³⁺@PUF on a lab coat via typical sewing operations, which can be used to monitor the exposure to X-ray. To better illustrate the application, we also incorporated the obtained HNTs@Na₅Lu₉F₃₂:Tb³⁺ into epoxy resin and fabricate it into drop shape pendant, which can also be used to indicate the exposure to X-ray.

The obtained HNTs@Na₅Lu₉F₃₂:Tb³⁺@PUF samples were cut to the required shapes (Fig. 2S) and then attached onto the lab coat via typical sewing operations (Fig. 2R). We also incorporate the obtained HNTs@Na₅Lu₉F₃₂:Tb³⁺ into an epoxy resin (Fig. 2P) and fabricate it into drop-shaped pendant with good transparency (Fig. 2Q). As shown in Fig. 2R, the obtained HNTs@Na₅Lu₉F₃₂:Tb³⁺@PUF-bearing lab coat and drop shape pendant are dressed on a full-length model. We applied the obtained materials in radiological laboratories. The results indicate that both the foams (Fig. 2T) and drop-shaped pendant (Fig. 2O) can emit visible green light under low dose X-ray (3.1 cGy/s).

The above-mentioned results and discussion have been added in Line 288-297 in the revised manuscript. For convenience, we have also added the images in the following:

Part of Fig. 3. Application of HNTs@Na₅Lu₉F₃₂:Tb³⁺-based composite in monitoring X-ray. **O** Photograph of the drop shape pendant taken under X-ray (3.1 cGy/s). **P** Photograph of the drop shape HNTs@Na₅Lu₉F₃₂:Tb³⁺@epoxy resin composite. **Q** Illustration of the transparency of the drop shape pendant. **R** Photograph of the mannequin wearing special-made lab coat and drop shape pendant. **S** Photograph of HNTs@Na₅Lu₉F₃₂:Tb³⁺@PUF fabricated on the lab coat. **T** Photograph of HNTs@Na₅Lu₉F₃₂:Tb³⁺@PUF taken under X-ray (3.1 cGy/s).

4. Flexible scintillator screens have the advantage of enabling high-quality imaging on

non-planar objects and offering a broader range of applications compared to rigid screens (*Adv. Mater.* 2022, 34, 2204801.). In this study, a rigid scintillator screen was prepared on a glass plate using a similar method as TLC plate preparation. It raises the question of whether the same procedure can be employed to prepare flexible polymer-based scintillator screens, allowing for mass production. Additionally, it is worth noting that the image quality achieved with scintillator screens in this work is considered mediocre.

Response: We thank the reviewer for this comment and have recognized the significance of the flexible scintillator after carefully reading the literature from *Adv. Mater.* 2022, 34, 2204801. The significance of the flexible scintillator is emphasized in Line 89-90 and Line 345-349 in the revised manuscript.

In the revised manuscript, we have included a facile method to prepare flexible scintillator screen by incorporating the obtained HNTs@Na₅Lu₉F₃₂:Tb³⁺ into a 1,4-phenyldiboronic acid-crosslinked hydrogel. In addition, we also made attempts to modify the surface of Na₅Lu₉F₃₂:Tb³⁺ with oleic acid (OA) to give the product OA@Na₅Lu₉F₃₂:Tb³⁺. A comparison between HNTs@Na₅Lu₉F₃₂:Tb³⁺- and OA@Na₅Lu₉F₃₂:Tb³⁺-incorporated hydrogels on tensile properties is conducted. The preparation method is added in the following:

The flexible screen is prepared following another method: 1.35 g PVA was dissolved in 12.0 mL distilled water at 80 °C. 65 mg of HNTs@Na₅Lu₉F₃₂:Tb³⁺ or OA@Na₅Lu₉F₃₂:Tb³⁺ was suspended in the PVA solution by intensely stirring. The solution was transferred into a specially-made rubber mold under argon, followed by the addition of 1,4-phenyldiboronic acid-containing NaOH solution (4.8×10⁻³ mmol / mL). The mixture was then introduced into a constant temperature oven and kept at 80°C for 1 h. After cooling to room temperature, the gel was picked out and then cut into the desired shapes to afford a flexible X-ray scintillator screen for imaging studies.

The major advantage of flexible X-ray scintillator screens is that they can be used for non-flat objects since they can be bended or stretched into desired shapes to accommodate non-flat objects (*Adv. Mater.* **34**, 2204801 (2022); *Mater. Horiz.* **7**, 1613-1622 (2020)). Therefore, the tensile strength is an important factor for designing flexible X-ray scintillator screens. The incorporation of OA@Na₅Lu₉F₃₂:Tb³⁺ gives rise to a lowered tensile strength for the obtained hydrogel. Anchoring Na₅Lu₉F₃₂:Tb³⁺ on the surface of halloysite nanotubes (HNTs) can effectively solve the problem. The HNTs@Na₅Lu₉F₃₂:Tb³⁺-containing hydrogel shows a much better tensile strength than the OA@Na₅Lu₉F₃₂:Tb³⁺-containing hydrogel, even higher than the pristine hydrogel, which is likely related to orientation of the HNTs during the stretching process, which is known to improve the tensile strength (*Prog. Polym. Sci.* 38, 1690-1719 (2013)).

Fig. S16 displays a series of photographs of the HNTs@Na₅Lu₉F₃₂:Tb³⁺-containing hydrogel under different tensile strains and bending angles. It can be bent into 90° without any breaking and the elongation can be increased from 0% to 250% without rupture. Moreover, the feasibility of X-ray imaging by the HNTs@Na₅Lu₉F₃₂:Tb³⁺-containing hydrogel is also demonstrated based on the “capsule” model (Fig.S16L). Anchoring nanoscintillators onto the surface of HNTs may

pave an alternative path in developing flexible scintillator materials with good mechanical properties.

The above-mentioned methods, results and discussion have been added in Line 505-512 and Line 345-367 in the revised manuscript.

The image achieved with scintillator screens in Fig. 4E has been improved in the revised manuscript. To achieve this, the HNTs@Na₅Lu₉F₃₂:Tb³⁺ powders were filtered by a 200-mesh sieve before the mixture with CMC-Na. The obtained screens were carefully polished. Additionally, the space between the “capsule” sample and X-ray source was adjusted to achieve an optimal image quality. Additionally, we also tried our best to improve the quality of graphical abstract and Scheme 1 in the revised manuscript.

For convenience, we have also added the newly revised Fig. 4, Fig. S15, Fig. S16, graphical abstract, and Scheme 1 in the following:

Fig. 4. X-ray imaging studies. **A** Scheme of the production and composition of the X-ray scintillator screen. **B** Photograph of the as-prepared X-ray scintillator screen (Thickness: ca.1 mm). **C** Fabrication of the “spring-containing capsule” model. **D** Schematic diagram of the X-ray imaging system based on the as-prepared X-ray scintillator screen. **E** The image relating to the capsule model taken from the digital camera (X-ray: 30 kV, 20 mA).

Fig. S15. Characterization of oleic acid-modified $\text{Na}_5\text{Lu}_9\text{F}_{32}:\text{Tb}^{3+}$ ($\text{OA}@\text{Na}_5\text{Lu}_9\text{F}_{32}:\text{Tb}^{3+}$). **A XPS pattern of $\text{OA}@\text{Na}_5\text{Lu}_9\text{F}_{32}:\text{Tb}^{3+}$. **B** Tb 3d region. **C** Lu 4d5 region. **D** C 1s region. **E** Photograph of $\text{OA}@\text{Na}_5\text{Lu}_9\text{F}_{32}:\text{Tb}^{3+}$ in water (1 mg/mL).**

Fig. S16. Fabrication and properties of flexible X-ray scintillator screen. **A** Preparation of HNTs@Na₅Lu₉F₃₂:Tb³⁺-incorporated hydrogel which was used as flexible X-ray scintillator screen in this study. **B-E** Photographs of flexible X-ray scintillator screen at different stretch lengths. **F-I** Photographs of flexible X-ray scintillator screen under different bending angles. **J** Stress-strain curves of tensile testing. **K** Schematic diagram of the X-ray imaging system based on the as-prepared flexible X-ray scintillator screen. **L** The image relating to the capsule model taken from the digital camera.

Water-dispersible halloysite nanotube X-ray scintillator

Graphical abstract

Scheme 1. Synthesis of HNT-CA and HNTs@Na₅Lu₉F₃₂:Tb³⁺.

5. In Section 2.4, the author uses HNTs@Na₅Lu₉F₃₂:Tb³⁺ to make multi-layer hydrogel for information encryption. In fact, the use of hydrogels for information encryption has been extensively studied, but I'm not sure what new benefits and breakthroughs this protocol provides.

Response: We thank the reviewer for raising this critical constructive comment. To more clearly demonstrate the added value of our materials for information encryption, we have redesigned the experiments by introducing the concept of “false information” under UV light. The revised preparation method is shown in Sec. 4.1.6 Page 16 in the revised manuscript. The results and discussion relating to the information encryption

studies have been also rewritten in Sec. 2.4 Page 13-14 in the revised manuscript. By utilizing the different emission behaviors under X-ray and UV light, anti-counterfeiting technology is introduced in the hydrogels for information encryption. Following this way, the X-ray-decrypted hydrogels can afford a securer way to protect the information.

For review's convenience, we have also added the detailed methods, results and discussion in the following:

“...4.1.6. HNTs@Na₅Lu₉F₃₂:Tb³⁺-containing hydrogel and multilayer hydrogels

1.35 g PVA was dissolved in 12.0 mL distilled water at 80 °C. 130 mg of HNTs@Na₅Lu₉F₃₂:Tb³⁺ was suspended into the PVA solution by intensely stirring. 1.5 mL of a M3-containing NaOH solution (4.8×10⁻³ mmol / mL, pH = 10.5) was added into this aqueous system. The system was stirred for about 5 min and then transferred to the tailor-made molds (letter-like). The samples were stored at -20 °C for 2 h and defrosted at room temperature for 1 h to afford Gel-1 as a self-standing hydrogel with a “letter” shape (“H”, “B” and “U”). Gel-2 was also prepared in letter-like molds following the similar procedure with the replacement of HNTs@Na₅Lu₉F₃₂:Tb³⁺ to pristine HNTs to give the hydrogels with shape for the letter "E" and “T”.

Gel-0 was prepared in a rectangular mold (7.5 cm × 2.5 cm) by using PVA and HNTs as substrate material. 1,4-phenyldiboronic acid was used as the crosslinking agent. To prepare the multilayer hydrogel, the Gel-0 was placed on the bottom of the rectangular mold. Then Gel-1 and Gel-2 samples were added onto Gel-0 piece following the sequence: E-H-B-U-T. Another layer of Gel-0 was prepared in-situ on the top of the layer containing Gel-1 and Gel-2 to give the hydrogel multilayer system. Then the overlaid samples were heated at 80 °C for 30 min to give the sandwich-like multilayer hydrogels. This process is summarized in Scheme 4...”

Scheme 4. Synthesis of Gel-0, Gel-1, and Gel-2 and illustration of different

emission behaviors under X-ray or UV.

“2.4. Information encryption in multilayer hydrogels...”

...It should be noted that some luminescent materials can exhibit different emission behavior between X-ray and UV light. The obtained HNTs@Na₅Lu₉F₃₂:Tb³⁺ can emit green light under X-ray. Some other organic materials can only generate luminescence under UV light while cannot emit light under X-ray. Such differences in emission behaviors inspired us to combine an anti-counterfeiting technology into the hybrid hydrogel with encrypted information. In our strategy, the “false information” can be read under UV light, while the “encrypted information” can only be recognized by X-ray. Following this way, the X-ray-decrypting hydrogels can afford a securer way to protect the information.

To implement this, three kinds of hydrogels were synthesized and the schematic diagram is shown in Scheme 4. Gel-0 is prepared by using 1,4-phenyldiboronic as the crosslinker and used as the substrate to prepare multilayer hydrogels, which cannot emit any luminescence under UV or X-ray. HNTs@Na₅Lu₉F₃₂:Tb³⁺-containing hydrogels (Gel-1) were prepared in letter-type molds. The boronic acid functionalized tetraphenylethylene (M3) is employed, which gives rise to the three-dimensional network in Gel-1. The presence of both tetraphenylethylene (TPE) units and HNTs@Na₅Lu₉F₃₂:Tb³⁺ makes Gel-1 exhibit a strong blue fluorescence due to the aggregation-induced emission (AIE) effect, and also endows an RL behavior to Gel-1. Gel-2 is prepared in a similar manner without the addition of HNTs@Na₅Lu₉F₃₂:Tb³⁺ making Gel-2 only photoluminescent (PL) exhibits. Gel-1 and Gel-2 were prepared in “letter” shapes to compose the “information layer”. The presence of boronic ester crosslinks and potential residual arylboronic acid in the hydrogels including Gel-0, Gel-1, and Gel-2 induce self-healing properties and dynamic exchange behavior. Herein, the design of a “sandwich”-like multilayer hydrogel with encrypted information is proposed (Fig. 5A), which consists of two outer non-luminescence layers (Gel-0) and an inner “information layer” (sequence: Gel-0/ “information layer” /Gel-0).”

“The obtained multilayer hydrogel cannot show any information under normal version. When exposed to UV light, the letters “EHBUT” with luminescence can be clearly observed in Fig. 5E. No differences can be found among the luminescence from the five letters. The information recognized from UV light is defined as “false information”. The encrypted letters (“HBU” shown in Fig.5F) can only be identified with the exposure to X-ray.

Therefore, the designed multilayer hydrogel allows encryption of specific information that can only be read out when exposed to X-ray irradiation while false information is seen under UV light irradiation providing a new way of data encryption. By utilizing the different emission behaviors under X-ray and UV light, anti-counterfeiting technology is introduced in the hydrogels for information encryption. Furthermore, more complicated information, including graphics, QR codes, and even 3D information, is also expected to be programmed into the multilayer hydrogel following this way.”

Fig. 5. Information encryption studies. **A** Scheme of preparation of multilayer hydrogels. **B** Normal version of the prepared multilayer hydrogel. **C** Bended hydrogel sample. **D** Twisted hydrogel sample. **E** Photograph of the prepared multilayer hydrogel under UV light (365 nm). **F** Photograph of the encrypted information “HBU” in the multilayer hydrogel after exposure to X-ray (30 kV, 20 mA).

Reviewer #3 (Remarks to the Author):

The manuscript presents the synthesis of $\text{Na}_5\text{Lu}_9\text{F}_{32}:\text{Tb}^{3+}$ nanoparticles (NPs) on surface modified-halloysite nanotubes, which are applied as X-ray scintillators in radioluminescent polymer materials, X-ray imaging coatings and information encrypting hydrogels.

The work presents novel results. It should be noted that $\text{Na}_5\text{Lu}_9\text{F}_{32}:\text{Tb}^{3+}$, while a known material, has not been previously reported as X-ray scintillator. Moreover, the deposition of X-ray scintillators on halloysite nanotubes is also here reported for the first time (even though composites of X-ray scintillators with SiO_2 have already been reported). Another noteworthy approach is the preparation of a multilayer hydrogels containing the synthesized composite material, which can be used for information encryption.

Overall, the manuscript could possibly be published pending major revisions.

Response: Thanks for your valuable comments and support of our work. Please see the detailed point-by-point responses in the following.

1. The significance of these approaches is more dubious. One of the main aims of the work, as described by the authors in the Introduction, is the development of “water-dispersible X-ray scintillators”. The solution here adopted is based on the deposition of the X-ray scintillator nanoparticles on a surface-modified clay material (citrate-APTES-grafted halloysite nanotubes, HNT). It is unclear, however, why the authors did not simply modify via covalent grafting with the same organic moieties the surface of the $\text{Na}_5\text{Lu}_9\text{F}_{32}:\text{Tb}^{3+}$ nanoparticles. Furthermore, the use of halloysite nanotubes is not properly justified. The authors claim that halloysite nanotubes are non-toxic, however contradicting literature studies have been reported (see e.g., *Toxicological Research* 37, 301–310 (2021)). The authors do not exploit the characteristic structural and morphological features of HNTs, since they simply deposit the NPs on the external surface of the nanotubes. Silica NPs or other clay minerals could have been used instead, especially considering that those materials are already widely used commercially as polymer fillers and in surface coatings.

Response: We thank the reviewer for sharing these important constructive critical remarks. There are indeed many surface modifications methods in the literature that focus on improving the water-dispersibility and biocompatibility of nanoscintillators. In these studies, oleic acid (OA), polyethylene glycol (PEG), and folic acid are usually employed to achieve the surface modification of nanoscintillators. We also made attempts to modify the surface of $\text{Na}_5\text{Lu}_9\text{F}_{32}:\text{Tb}^{3+}$ by oleic acid to give the product $\text{OA}@\text{Na}_5\text{Lu}_9\text{F}_{32}:\text{Tb}^{3+}$ as confirmed by XPS (Fig. S15). The water-dispersibility of the nanoscintillators can be improved in some content, but after being incorporated into the hydrogel, it gives rise to a lowered tensile strength for the obtained hydrogel. Anchoring $\text{Na}_5\text{Lu}_9\text{F}_{32}:\text{Tb}^{3+}$ on the surface of halloysite nanotubes (HNTs) can effectively solve the

problem. The HNTs@Na₅Lu₉F₃₂:Tb³⁺-containing hydrogel shows a much better tensile strength than the OA@Na₅Lu₉F₃₂:Tb³⁺-containing hydrogel, even higher than the pristine hydrogel, which is likely related to orientation of the HNTs during the stretching process, which is known to improve the tensile strength (*Prog. Polym. Sci.* 38, 1690-1719 (2013)). The major advantage of flexible X-ray scintillator screens is that they can be used for non-flat objects since they can be bended or stretched into desired shapes to accommodate non-flat objects (*Adv. Mater.* 34, 2204801 (2022); *Mater. Horiz.* 7, 1613-1622 (2020)). Therefore, the tensile strength is an important factor for designing flexible X-ray scintillator screens. Comparatively, HNTs@Na₅Lu₉F₃₂:Tb³⁺-containing hydrogel should have better application foreground in X-ray scintillator screens than OA@Na₅Lu₉F₃₂:Tb³⁺-incorporated one. Following this way, the characteristic structural and morphological features of HNTs are well exploited. That is also the reason why we used HNTs rather than silica NPs or other clay minerals as substrate.

Fig. S16 displays a series of photographs of the HNTs@Na₅Lu₉F₃₂:Tb³⁺-containing hydrogel under different tensile strains and bending angles. It can be bent to 90° without any breaking and the elongation can be increased from 0% to 250% without rupture. Moreover, the feasibility of X-ray imaging by the HNTs@Na₅Lu₉F₃₂:Tb³⁺-containing hydrogel is also demonstrated based on the “capsule” model (Fig. S16L). Anchoring nanoscintillators onto the surface of HNTs may pave an alternative path in developing flexible scintillator materials with good mechanical properties.

The above-mentioned content has been added in Line 86-90 and Line 345-367 in the revised manuscript. Detailed results have been added in Fig. S15 and S16 in the revised supplementary information. For review’s convenience, we have also added Fig. S15 and S16 in the following:

Fig. S15. Characterization of oleic acid-modified $\text{Na}_5\text{Lu}_9\text{F}_{32}:\text{Tb}^{3+}$ ($\text{OA}@\text{Na}_5\text{Lu}_9\text{F}_{32}:\text{Tb}^{3+}$). **A** XPS pattern of $\text{OA}@\text{Na}_5\text{Lu}_9\text{F}_{32}:\text{Tb}^{3+}$. **B** Tb 3d region. **C** Lu 4d5 region. **D** C 1s region. **E** Photograph of $\text{OA}@\text{Na}_5\text{Lu}_9\text{F}_{32}:\text{Tb}^{3+}$ in water (1 mg/mL).

Fig. S16. Fabrication and properties of flexible X-ray scintillator screen. **A** Preparation of $\text{HNTs}@\text{Na}_5\text{Lu}_9\text{F}_{32}:\text{Tb}^{3+}$ -incorporated hydrogel which was used as flexible X-ray scintillator screen in this study. **B-E** Photographs of flexible X-ray scintillator screen at different stretch lengths. **F-I** Photographs of flexible X-ray scintillator screen under different bending angles. **J** Stress-strain curves of tensile testing. **K** Schematic diagram of the X-ray imaging system based on the as-prepared flexible X-ray scintillator screen. **L** The image relating to the capsule model taken from the digital camera.

For a deep learning about the toxicity of HNTs, we have downloaded the literature from Toxicological Research 37, 301–310 (2021). The authors demonstrated the

potential toxicity of HNTs on human alveolar carcinoma epithelial cells (A549) and human bronchial epithelial cells (BEAS-2B). We had overlooked this study, because numerous research articles and reviews hold the view that HNTs are non-toxic. We have changed the “non-toxic” to “low toxicity” in the revised manuscript. Because the prepared HNTs@Na₅Lu₉F₃₂:Tb³⁺ was not subjected to medical uses in vivo, the potential toxicity will not be discussed in-depth in our study. We also quoted this reference in Line 80 Introduction part in revised manuscript, to remind the readers that still some controversy on the potential toxicity of HNTs: “...Though there is still some controversy on the potential toxicity upon some special items (*Toxicol. Res.* **37**, 301-310 (2021)), HNTs remain a popular topic in material science including biomedical uses with a growing trend of attention (*Nat. Commun.* **14**, 5083 (2023))...”

2. The authors present extensive characterization data, particularly concerning the structural and morphological properties of the HNTs@Na₅Lu₉F₃₂:Tb³⁺. However, the authors claimed that “Na₅Lu₉F₃₂:Tb³⁺ is introduced as new X-ray scintillating material” but the characterization of the main performance parameters of this new scintillator lacks important data (e.g., conversion efficiency, decay time, PL emission). Furthermore, the stability of the composite material with polyurethane foam and of the X-ray scintillator screen should be investigated in a more quantitative fashion.

Response: We thank the reviewer for pointing out this lack of characterization for the new X-ray scintillator material. We have performed additional characterizations of Na₅Lu₉F₃₂:Tb³⁺, including XPS, XRD analysis, SEM observations, RL emission spectrum and decay time. Moreover, we also conducted a series of characterization of NaLuF₄:Tb³⁺. The synthesis method is added in Sec. S1.4 in revised supplementary information. A systematical comparison between Na₅Lu₉F₃₂:Tb³⁺ and NaLuF₄:Tb³⁺ was conducted to reveal the differences in structure and emission properties. Details have been added in Sec. S2.2, Fig. S1 and Fig. S2 in revised supplementary information. For convenience, we have also added them in the following:

“Na₅Lu₉F₃₂:Tb³⁺ and NaLuF₄:Tb³⁺ were synthesized via a solvothermal method by using different chelating agent. The peaks in X-ray diffraction (XRD) patterns of Na₅Lu₉F₃₂:Tb³⁺ and NaLuF₄:Tb³⁺ shown in Fig. S1A and Fig. S2A match well with the standard Na₅Lu₉F₃₂ (JCPDS No. 27-0725) and hexagonal phase NaLuF₄ (JCPDS No. 27-0726), respectively. The XRD results indicate that the obtained Na₅Lu₉F₃₂:Tb³⁺ obeys to the character of cubic lattice system ($a = b = c = 5.464 \text{ \AA}$) and the density is calculated as 6.14 g/cm^3 .

SEM and TEM were used to reveal the micromorphology characters. The results show that the obtained Na₅Lu₉F₃₂:Tb³⁺ exhibits unregular sphere-like particles. For NaLuF₄:Tb³⁺, regular hexagonal phase can be obtained. XPS was used to reveal the chemical composition of Na₅Lu₉F₃₂:Tb³⁺ and NaLuF₄:Tb³⁺. The presence of sodium, lutetium, fluorine, and terbium is demonstrated based on the peaks around 1277 eV (Tb 3d3), 1243 eV (Tb 3d4), 1072 eV (Na 1s), 686 eV (F 1s), and 198 eV (Lu 4d5), respectively. Both of the RL spectra of Na₅Lu₉F₃₂:Tb³⁺ and NaLuF₄:Tb³⁺ features four

emission peaks at 489, 544, 585 and 620 nm. The X-ray-induced long persistent luminescence properties were investigated. The persistent luminescence decay curves monitored at 544 nm after irradiation by an X-ray irradiator for 10 min were recorded, as shown in Fig. S2H. The afterglow intensity decreased quickly in the first hour and then decayed slowly. Even after 4×10^3 s, the afterglow can even be detected. $\text{Na}_5\text{Lu}_9\text{F}_{32}:\text{Tb}^{3+}$ cannot any persistent luminescence behaviors. The decay time of $\text{Na}_5\text{Lu}_9\text{F}_{32}:\text{Tb}^{3+}$ is found less than 1 s, which exhibits a synchronous RL behavior following the “On-Off” switching of X-ray irradiation (Fig. S1H).”

Fig. S1. Structural characterizations and optical properties of $\text{Na}_5\text{Lu}_9\text{F}_{32}:\text{Tb}^{3+}$. **A** XRD pattern of $\text{Na}_5\text{Lu}_9\text{F}_{32}:\text{Tb}^{3+}$ and the comparison with standard. **B** XPS spectrum of $\text{Na}_5\text{Lu}_9\text{F}_{32}:\text{Tb}^{3+}$. **C-F** TEM images of $\text{Na}_5\text{Lu}_9\text{F}_{32}:\text{Tb}^{3+}$. **G** X-ray-excited radioluminescence (RL) spectrum of $\text{Na}_5\text{Lu}_9\text{F}_{32}:\text{Tb}^{3+}$. **H** In situ measurement of the luminescence intensity of $\text{HNTs}@\text{Na}_5\text{Lu}_9\text{F}_{32}:\text{Tb}^{3+}$ under X-ray with “On-Off” cycles. **I** SEM image of $\text{Na}_5\text{Lu}_9\text{F}_{32}:\text{Tb}^{3+}$.

Fig. S2. Structural characterizations and optical properties of NaLuF₄:Tb³⁺. **A** XRD pattern of NaLuF₄:Tb³⁺ and the comparison with standard. **B** XPS spectrum of NaLuF₄:Tb³⁺. **C&D** SEM images of NaLuF₄:Tb³⁺. **E** X-ray-excited luminescence emission spectrum and afterglow spectra of NaLuF₄:Tb³⁺. **F&G** TEM images of NaLuF₄:Tb³⁺. **H** Afterglow intensity from NaLuF₄:Tb³⁺ monitored at 544 nm as a function of time.

A WD-2A stability test instrument was used to evaluate the stability of HNTs@Na₅Lu₉F₃₂:Tb³⁺@PUF on the X-ray scintillator screen (Temperature: 60 ± 0.5°C; Light intensity: 2000 lx, white; Humidity: 50 ± 4%). The samples were placed in the stability test instrument for 30 days to evaluate the stability to heat and light stability. The X-ray-excited RL emission spectra of the samples before and after stability test were recorded. After the stability test, the RL emission spectra of HNTs@Na₅Lu₉F₃₂:Tb³⁺@PUF and X-ray scintillator screen is not significantly changed compared to the original samples. No newly appeared peaks were detected and the peak intensity of 544 nm was retained for more than 95%. The results indicate the prepared HNTs@Na₅Lu₉F₃₂:Tb³⁺@PUF and X-ray scintillator screen exhibit good stability to heat and good light stability.

The results have been added in Line 264 and 327 in the revised manuscript and Fig. S13 in revised supplementary information. For convenience, we also add Fig. S13 in the following.

Fig. S13. X-ray-excited RL emission spectra of the samples before and after stability test. A HNTs@Na₅Lu₉F₃₂:Tb³⁺@PUF. **B** HNTs@Na₅Lu₉F₃₂:Tb³⁺-based rigid X-ray scintillator screen.

3. Data discussion could be improved. The image of a single functionalized nanotube is reported: this is not sufficient to prove the homogeneity of the sample. No estimate about the average size of the NPs is reported. Discussion of several characterization techniques (XPS, XRD, TGA...) is only qualitative. Furthermore, some details should be revised. For instance, in solid state NMR spectra, it makes no sense to report signal position with two decimal digits. In Figure S7, please double check the scale bars (the panel on the right seems to report a larger magnification, but the scale bar does not support that).

Response: We thank the reviewer for pointing out these shortcomings, which have been addressed during revision. We have performed dynamic light scattering (DLS; measurement by using a commercialized spectrometer from Brookhaven BI-200SM Goniometer equipping a 17 mW He–Ne laser (633 nm); A Laplace inversion program was used to process the data to obtain the effective diameter and polydispersity index (PDI)) revealing an effective diameter of the obtained HNTs@Na₅Lu₉F₃₂:Tb³⁺ of 503 nm with a polydispersity index PDI of 0.150, suggesting a good homogeneity of the sample. The results have been added in Fig. S10 in revised manuscript. For convenience, we also add the distribution histogram in the following:

Fig. S10B. Size distributions of HNTs@Na₅Lu₉F₃₂:Tb³⁺ in aqueous solution.

To add some qualitative data in our study, we performed X-ray analysis of Na₅Lu₉F₃₂:Tb³⁺. The obtained Na₅Lu₉F₃₂:Tb³⁺ obeys to the character of cubic lattice system ($a = b = c = 5.464 \text{ \AA}$) and the density is calculated as 6.14 g/cm^3 .

In the solid-state ¹³C NMR spectrum, the signal position is marked with one decimal digit referring to other HNTs-related studies containing solid-state ¹³C NMR characterizations (*J. Am. Chem. Soc.* **134**, 12134-12137 (2012)). For convenience, we also add the revised spectra in the following:

Fig. S4. Solid-state ¹³C NMR spectrum and ¹³C NMR spectrum (DMSO-*d*₆) of

CA.

We have checked the images in Fig. S7 (now numbered as Fig. S8 in revised supplementary information), which indeed had a mistake in the scale bars. To address this comment, we performed additional TEM images, and now the scale bars match well with the magnifications. The tubular structures are also demonstrated. Moreover, the elemental mapping analysis is also conducted to verify the chemical composition of pristine HNTs. O, Si, and Al are found to be distributed along the nanotube, which is in accordance with well-accepted compositions of HNTs.

For convenience, we also add the TEM results in the following:

Fig. S8. TEM results of pristine HNTs. A-D TEM images. **E** High-angle annular dark-field STEM image. **F** Oxygen. **G** Silicon. **H** Aluminum.

Reviewer #4 (Remarks to the Author):

This paper reports on a new nanoparticle X-ray scintillator: HNTs@Na₅Lu₉F₃₂:Tb³⁺ prepared by hydrothermal synthesis. The radioluminescent Na₅Lu₉F₃₂:Tb³⁺ nanoparticles were decorated on HNTs nanotubes and incorporated in three different composite platforms for potential scintillation applications. A high light yield of 15800 photons MeV⁻¹ is reported for Na₅Lu₉F₃₂:Tb³⁺ nanoparticles. The high water dispersibility of the particles facilitates its processing in organic or inorganic matrices for large scale manufacturing. The new material seems promising and its versatile processing capability is interesting for X-ray sensing or imaging applications.

Response: We thank the reviewer for the valuable comments and support of our work. Please see the detail point-by-point responses in the following.

1. It is not clear how the X-ray attenuation coefficient for the scintillator and LuAG:Ce scintillator are determined for the calculation of the light yield. Is Na₅Lu₉F₃₂:Tb³⁺ material only considered for attenuation coefficient calculation or the whole screen scintillator? Information on sample thickness and volume, density, X-ray energy etc. should be provided.

Response: We thank the reviewer for pointing out this unclarity and we have resolved this in the revised manuscript. Na₅Lu₉F₃₂:Tb³⁺ was not subject to the screen scintillator studies. The X-ray attenuation coefficient was calculated by measuring I_0 and I (X-ray intensities of the incident and transmitted beams). It is possible to determine the X-ray attenuation coefficient a as:

$$a = -\frac{1}{x} \ln \frac{I}{I_0} \quad (\text{S2})$$

where x is the thickness of the sample. I_0 and I represent the X-ray intensities of the incident and transmitted beams, respectively (*Radiat. Phys. Chem.* **75**, 936-944 (2006)).

The volume and thickness for all samples (cylindrical in shape) were 0.0157 cm³ and 0.2 mm, respectively. The density of standard sample (LuAG:Ce), Na₅Lu₉F₃₂:Tb³⁺, and HNTs@Na₅Lu₉F₃₂:Tb³⁺ were calculated as 6.72, 3.76 and 3.63 g/cm³, respectively. A 30 mA 40 kV X-ray irradiator is used in measuring the X-ray attenuation coefficient and light yields.

The above-mentioned method and parameters have been added in Sec. S1.5 in the revised supplementary information.

2. A more detailed characterization on the structure and properties of Na₅Lu₉F₃₂:Tb³⁺ nanoparticle scintillator should be presented. For example, the crystalline structure, density, decay time etc. And a comparison with other nanoparticle scintillators (perovskites, NaLuF₄:Tb³⁺ etc) reported should be included. Does any persistent

radioluminescence as in NaLuF₄:Tb³⁺ observed from this new structure?

Response: This is a nice suggestion and we have performed additional characterizations of Na₅Lu₉F₃₂:Tb³⁺, including XPS, SEM observations, RL emission spectrum, decay time and analysis of crystalline structure. Moreover, we also conducted a serious of characterization of NaLuF₄:Tb³⁺. Persistent radioluminescence behavior was demonstrated. A systematical comparison between Na₅Lu₉F₃₂:Tb³⁺ and NaLuF₄:Tb³⁺ was conducted to reveal the differences in structure and emission properties. Details have been added in Sec. S2.2, Fig. S1 and Fig. S2 in the revised supplementary information. For convenience, we also add them in the following:

“Na₅Lu₉F₃₂:Tb³⁺ and NaLuF₄:Tb³⁺ were synthesized via a solvothermal method by using different chelating agent. The peaks in X-ray diffraction (XRD) patterns of Na₅Lu₉F₃₂:Tb³⁺ and NaLuF₄:Tb³⁺ shown in Fig. S1A and Fig. S2A match well with the standard Na₅Lu₉F₃₂ (JCPDS No. 27-0725) and hexagonal phase NaLuF₄ (JCPDS No. 27-0726), respectively. The XRD results indicate that the obtained Na₅Lu₉F₃₂:Tb³⁺ obeys to the character of cubic lattice system ($a = b = c = 5.464 \text{ \AA}$) and the density is calculated as 6.14 g/cm^3 .

SEM and TEM were used to reveal the micromorphology characters. The results show that the obtained Na₅Lu₉F₃₂:Tb³⁺ exhibits unregular sphere-like particles. For NaLuF₄:Tb³⁺, regular hexagonal phase can be obtained. XPS was used to reveal the chemical composition of Na₅Lu₉F₃₂:Tb³⁺ and NaLuF₄:Tb³⁺. The presence of sodium, lutetium, fluorine, and terbium is demonstrated based on the peaks around 1277 eV (Tb 3d₃), 1243 eV (Tb 3d₄), 1072 eV (Na 1s), 686 eV (F 1s), and 198 eV (Lu 4d₅), respectively. Both of the RL spectra of Na₅Lu₉F₃₂:Tb³⁺ and NaLuF₄:Tb³⁺ features four emission peaks at 489, 544, 585, and 620 nm. The X-ray-induced long persistent luminescence properties were investigated. The persistent luminescence decay curves monitored at 544 nm after irradiation by an X-ray irradiator for 10 min were recorded, as shown in Fig. S2H. The afterglow intensity decreased quickly in the first hour and then decayed slowly. Even after 4×10^3 s, the afterglow can even be detected. Na₅Lu₉F₃₂:Tb³⁺ cannot any persistent luminescence behaviors. The decay time of Na₅Lu₉F₃₂:Tb³⁺ is found less than 1 s, which exhibits a synchronous RL behavior following the “On-Off” switching of X-ray irradiation (Fig. S1H).”

Fig. S1. Structural characterizations and optical properties of $\text{Na}_5\text{Lu}_9\text{F}_{32}:\text{Tb}^{3+}$. **A** XRD pattern of $\text{Na}_5\text{Lu}_9\text{F}_{32}:\text{Tb}^{3+}$ and the comparison with standard. **B** XPS spectrum of $\text{Na}_5\text{Lu}_9\text{F}_{32}:\text{Tb}^{3+}$. **C-F** TEM images of $\text{Na}_5\text{Lu}_9\text{F}_{32}:\text{Tb}^{3+}$. **G** X-ray-excited radioluminescence (RL) spectrum of $\text{Na}_5\text{Lu}_9\text{F}_{32}:\text{Tb}^{3+}$. **H** In situ measurement of the luminescence intensity of HNTs@ $\text{Na}_5\text{Lu}_9\text{F}_{32}:\text{Tb}^{3+}$ under X-ray with “On-Off” cycles. **I** SEM image of $\text{Na}_5\text{Lu}_9\text{F}_{32}:\text{Tb}^{3+}$.

Fig. S2. Structural characterizations and optical properties of $\text{NaLuF}_4:\text{Tb}^{3+}$. **A** XRD pattern of $\text{NaLuF}_4:\text{Tb}^{3+}$ and the comparison with standard. **B** XPS spectrum of $\text{NaLuF}_4:\text{Tb}^{3+}$. **C & D** SEM images of $\text{NaLuF}_4:\text{Tb}^{3+}$. **E** X-ray-excited luminescence emission spectrum and afterglow spectra of $\text{NaLuF}_4:\text{Tb}^{3+}$. **F & G** TEM images of $\text{NaLuF}_4:\text{Tb}^{3+}$. **H** Afterglow intensity from $\text{NaLuF}_4:\text{Tb}^{3+}$ monitored at 544 nm as a function of time.

3. What is the X-ray energy used for X-ray imaging studies in Figure 4? What's the thickness of screen?

Response: An X-ray tube (30 kV, 20 mA, Power: 600 W) with W anode was used as the excitation source in Fig. 4. The thickness of screen is ca.1 mm. The information has been added in the figure captions.

4. Define “low-dose X-ray” that is used several times in the manuscript. What is the X-ray energy and what is the dosage.

Response: The “low-dose X-ray” used in this study means a 30 kV, 20 mA X-ray. The dose rate is 3.1 cGy/s which is much lower than that used in brachytherapy in clinical (*Int J Radiat Oncol* **72**, 665-670 (2008); *Clin Transl Radiat Oncol* **24**, 92-98 (2020)). We have clarified this point in Line 296 in revised manuscript.

5. What is the average size of $\text{Na}_5\text{Lu}_9\text{F}_{32}:\text{Tb}^{3+}$ nanoparticles and HNTs@ $\text{Na}_5\text{Lu}_9\text{F}_{32}:\text{Tb}^{3+}$. One advantage of nanoparticle scintillator is low scattering and the potential for the formation of transparent composite scintillator. Could the authors comment on the transparency of screen or other composites structures prepared to maximize light output?

Response: We performed the dynamic light scattering (DLS; measurement by using a commercialized spectrometer from Brookhaven BI-200SM Goniometer equipping a 17 mW He–Ne laser (633 nm); Laplace inversion program was used to process the data to obtain the effective diameter and polydispersity index (PDI)) revealing an effective diameter of the obtained HNTs@ $\text{Na}_5\text{Lu}_9\text{F}_{32}:\text{Tb}^{3+}$ is determined as 503 nm with a PDI of 0.15. For $\text{Na}_5\text{Lu}_9\text{F}_{32}:\text{Tb}^{3+}$, it shows an effective diameter of 520 with a PDI of 0.083. It should be noted that the diameter measured by the TEM observation of $\text{Na}_5\text{Lu}_9\text{F}_{32}:\text{Tb}^{3+}$ is ca. 20~30 nm. The much higher effective diameter calculated by the DLS can be attributed to the non-spherical character of the HNTs and potential aggregation of the $\text{Na}_5\text{Lu}_9\text{F}_{32}:\text{Tb}^{3+}$ nanoparticles. Details have been added in Fig. S10 in the revised supplementary information. For convenience, we also add them in the following:

Fig. S10. Size distribution in aqueous solution. **A** HNTs: effective diameter: 244 nm; PDI: 0.10. **B** HNTs@Na₅Lu₉F₃₂:Tb³⁺: effective diameter: 503 nm; PDI: 0.15. **C** Na₅Lu₉F₃₂:Tb³⁺: Effective diameter: 520 nm; PDI: 0.083.

For the objective of formation of transparent composite scintillator, we prepared the HNTs@Na₅Lu₉F₃₂:Tb³⁺@epoxy resin composite. The preparation method is shown in the following: 20 mg HNTs@Na₅Lu₉F₃₂:Tb³⁺ were dispersed in 10 mL of anhydrous ethanol and then added into the E51 solution (2 g in anhydrous ethanol). The mixture was sonicated for 5 min and then stirred at 70 °C for 2 h to obtain a well-dispersed suspension. The anhydrous ethanol was removed by rotary evaporation. The sample was weighted and the polyamide curing agent was added to the solution with a weight ratio of 1:1. The system was mixed by thorough agitation and then poured into the drop-shaped mold. The liquid-filled mold was degassed in a vacuum oven for 10 min and then placed at room temperature for 24 h to afford the HNTs@Na₅Lu₉F₃₂:Tb³⁺@epoxy resin composite. The obtained drop shape pendant shows good transparency and can emit visible green light under low dose X-ray (3.1 cGy/s). The above-mentioned preparation method and results have been added in Line 487, Line 290, and Fig. 3 in the revised manuscript. For convenience, we also add the images in the following:

Part of Fig. 3. Application of HNTs@Na₅Lu₉F₃₂:Tb³⁺-based composite in monitoring X-ray. **O** Photograph of the drop shape pendant taken under X-ray (3.1 cGy/s). **P** Photograph of the drop shape HNTs@Na₅Lu₉F₃₂:Tb³⁺@epoxy resin composite. **Q** Illustration of the transparency of the drop shape pendant. **R** Photograph of the mannequin wearing special-made lab coat and drop shape pendant.

REVIEWER COMMENTS

Reviewer #1 (Remarks to the Author):

The authors have addressed my concerns. This paper could be published in its present form.

Reviewer #3 (Remarks to the Author):

The manuscript has been revised thoroughly, convincingly addressing the raised points. I therefore recommend its publication in its present form.

Reviewer #5 (Remarks to the Author):

In this manuscript, the authors demonstrated the potential of a type of water-dispersible halloysite nanotube-based X-ray-sensitive scintillator with three kinds of developed targeting different applications for radioluminescent, X-ray imaging, and information encrypting materials. However, not enough scientific reasoning was provided to interpret and support the experimental data, so this manuscript is not recommended for publication in a prestigious journal such as Nature Communications. Please refer to the detailed comments below for consideration.

1. What are the reasons/mechanisms for the difference between the persistent radioluminescence from NaLuF₄:Tb³⁺ and synchronous RL behavior from Na₅Lu₉F₃₂:Tb³⁺ following the "On-Off" switching of X-ray irradiation while both compounds bear the same elements in different ratios? Why Na₅Lu₉F₃₂:Tb³⁺ does not possess persistent RL?
2. Related Scheme 2 and Figure 3, how deep could HNTs@Na₅Lu₉F₃₂:Tb³⁺ penetrate into PUF? Is it only "a uniform coating of HNTs@Na₅Lu₉F₃₂:Tb³⁺ on the PUF surface"? Experimental data should be provided, e.g. using cross-section SEM images and elemental mapping.
3. The authors need to take measurements to get key parameters of this new type of X-ray scintillators and compare them with what have been reported in the literature. To keep readers better informed, it is suggested to compile a table and include in the ESI.
4. What are the reasons/mechanisms for "some luminescent materials can exhibit different emission behavior between X-ray and UV light"? Are there any prior reports on this type of luminescent materials? If so, please list them in a table in ESI with adequate citations. And what are the advantages of the multilayer hydrogels reported in this manuscript?
5. To keep readers well informed, the authors are suggested to provide citations related to "Some other organic materials can only generate luminescence under UV light and cannot emit light under X-ray irradiation."
6. The authors mentioned that "The toxicity of lead is another severe issue restricting wide spread applications of perovskite-based X-ray scintillators" in the Introduction. However, many reported studies on perovskite-based X-ray scintillators do not contain lead, for

example, Ref. 8,9 and 11 used lead-free perovskites. The authors should compare their results with what have been reported before this manuscript could be validated for publication on Nature Communications.

7. By the way, the title reads too long. It should be shortened significantly to signify the key message of the work.

8. There are typos and grammatical errors to be corrected. For example,
Line 100, page 4: "In contrast tot the persistent radioluminescence"
Line 331, page 12: "the here developed lead-free composite"

Reviewer #1 (Remarks to the Author):

The authors have addressed my concerns. This paper could be published in its present form.

Response: Thanks a lot for your valuable comments during the first revision, which allowed us to greatly improve the quality of the manuscript.

Reviewer #3 (Remarks to the Author):

The manuscript has been revised thoroughly, convincingly addressing the raised points. I therefore recommend its publication in its present form.

Response: Thanks a lot for your valuable comments during the first revision, which allowed us to greatly improve the quality of the manuscript.

Reviewer #5 (Remarks to the Author):

In this manuscript, the authors demonstrated the potential of a type of water-dispersible halloysite nanotube-based X-ray-sensitive scintillator with three kinds of developed targeting different applications for radioluminescent, X-ray imaging, and information encrypting materials. However, not enough scientific reasoning was provided to interpret and support the experimental data, so this manuscript is not recommended for publication in a prestigious journal such as Nature Communications. Please refer to the detailed comments below for consideration.

Response: Thanks a lot for your valuable comments. To support enough scientific reasoning to better interpret the experimental data and strengthen the conclusion, we have performed significant additional work to explore the mechanisms for the different optical properties, to reveal the penetration depth of the composite into PUF, and to investigate the biocompatibility of the obtained HNTs@Na₅Lu₉F₃₂:Tb³⁺. Moreover, we have made systematic comparisons between this work and previous literature. Accordingly, the advantages of the obtained HNTs@Na₅Lu₉F₃₂:Tb³⁺ reported in this work and the advantages of using multilayer hydrogels were also summarized. Please see the point-by-point responses in the following:

1. What are the reasons/mechanisms for the difference between the persistent radioluminescence from NaLuF₄:Tb³⁺ and synchronous RL behavior from Na₅Lu₉F₃₂:Tb³⁺ following the “On-Off” switching of X-ray irradiation while both compounds bear the same elements in different ratios? Why Na₅Lu₉F₃₂:Tb³⁺ does not possess persistent RL?

Response: This is a very important and interesting comment. The mechanism of the X-ray-induced persistent luminescence phenomenon, also known as afterglow, is still under investigation and under debate. Up to now, several underlying mechanisms have been put forward to explain X-ray-induced persistent luminescence, including the hole trapping-detrapping model, the electron trapping-detrapping model, and the quantum tunneling model. (*Chem. Soc. Rev.* **45**, 2090-2136 (2016); *J. Lumin.* **205**, 581-620 (2019); *J. Mater. Chem. C* **6**, 6240-6249 (2018)) Although these models could explain some observed phenomena, there are flaws in these models and some key points still remain unclear.

To the best of our knowledge, persistent luminescence is decided by the following aspects: whether the excitation can effectively charge energy, whether the heat can effectively release charge carriers, and whether the luminescent center can effectively bind charge carriers and produce emissions. (*Nat. Nanotech.* **16**, 1011-1018 (2021); *J. Phys. Chem. C* **124**, 24940-24948 (2020)) Previous research has mainly concentrated on the former two aspects, while rarely researchers focused on the third topic. Conventionally, the binding ability between lanthanide ions and charge carriers is viewed to be affected by the inherent arrangement of the electrons and ionization energies of lanthanides. (*J. Electrochem. Soc.* **152**, H107 (2005); *Phys. Status. Solidi B Basic. Res.* **242**, R7-R9 (2005)) Ce³⁺, Pr³⁺, and Dy³⁺ can more easily bind traps, while Eu³⁺, Yb³⁺, Sm³⁺, and Tm³⁺ are more likely to bind electrons. The persistent luminescent abilities from the above lanthanides can be explained by this hypothesis. However, the persistent luminescence from Gd³⁺ can not be explained by this model because Gd³⁺ can not easily bind electrons nor traps. The Gd³⁺ doped ScPO₄ shows obvious X-ray-induced persistent luminescence, while the persistent luminescence ability seriously declines for Gd³⁺ doped YPO₄ and LuPO₄. The persistent luminescence is difficult to be detected in Gd³⁺ doped LaPO₄ with the same doping concentration. (*Light Sci. Appl.* **11**, 51 (2022)) The different persistent luminescence behavior in different hosts raised our concern and inspired us to explore alternative models.

To break the limitation, we proposed a mechanism for the trivalent lanthanides' persistent luminescence based on abundant experiments and analysis in a previous study. (*Light Sci. Appl.* **11**, 51 (2022)) According to the mechanism, the trivalent lanthanides as isoelectronic traps are expected to eventually bind excitons, and this binding ability is not only related to the inherent arrangement of the electrons of the trivalent lanthanides, but also to the extrinsic anion coordination and cation substitution in the host lattices. Following this way, the persistent luminescence that came from Gd³⁺ can be well explained and the persistent luminescent ability can be regulated by

changing the coordinated anions and substituted cations in the host lattices. The excitons in such materials transfer their recombination energy to the trivalent lanthanides, followed by the generation of persistent luminescence from the trivalent lanthanides.

In this study, X-ray is used for excitation, which can charge energy to nearly all kinds of hosts. The factors that govern the persistent luminescence properties and eventually lead to the presence or absence of afterglows may be attributed to the following aspects:

- i) character of the luminescent center;
- ii) character of hosts (coordinated anions, substituted cations, type of coordinated linkage, symmetry, etc.);
- iii) doping concentration of lanthanides.

$\text{Na}_5\text{Lu}_9\text{F}_{32}:\text{Tb}^{3+}$ and $\text{NaLuF}_4:\text{Tb}^{3+}$ have the same luminescent center and therefore display identically the same emission peaks in X-ray-excited radioluminescence spectra. To further evaluate the influence of the doping concentration of Tb^{3+} on the persistent luminescence properties, we prepared $\text{NaLuF}_4:\text{Tb}^{3+}$ with different doping concentrations of Tb^{3+} (0.1%, 1%, 5%, 10%, and 15%). The afterglow intensity from $\text{NaLuF}_4:\text{Tb}^{3+}$ monitored at 544 nm was also recorded as shown in Supplementary Fig. 18. All of the samples display typical afterglow curves and the optimal afterglow property can be found in the case of 10%. The results suggest that the persistent luminescence properties can be affected by the doping concentration of Tb^{3+} , while it is not the crucial factor to decide the presence or absence of afterglow. Therefore, the difference between the persistent radioluminescence from $\text{NaLuF}_4:\text{Tb}^{3+}$ and synchronous radioluminescence behavior from $\text{Na}_5\text{Lu}_9\text{F}_{32}:\text{Tb}^{3+}$ following the “On-Off” switching of X-ray irradiation should be attributed to the difference in the structure of the hosts.

Supplementary Figure 18. Afterglow intensity from $\text{NaLuF}_4:\text{Tb}^{3+}$ with different doping concentration of Tb^{3+} monitored at 544 nm as a function of time.

$\text{Na}_5\text{Lu}_9\text{F}_{32}$ and NaLuF_4 share the same elements. The major difference between them is the lattice types. According to the XRD results, $\text{Na}_5\text{Lu}_9\text{F}_{32}$ obeys the character of a cubic lattice system, while the lattice type of NaLuF_4 should be classified into a hexagonal phase. It has been reported that the difference between cubic and hexagonal lattice within the crystals bearing the same elements usually possess different binding energies, (*J. Phys. Condens. Matter* **20**, 075233 (2008); *AIP Adv.* **12**, 025330 (2022)) which eventually gives rise to different optical properties. (*Phys. Rev. B* **68**, 035312 (2003); *Chem. Mater.* **26**, 1881-1888 (2014)) Liu's group has demonstrated that the hexagonal-phased lattices composed of Na, Ln (lanthanide), and F are more suitable for achieving energy transfer and energy migration, as opposed to the cubic-phased counterpart. (*Nat. Commun.* **8**, 899 (2017)) On the other hand, the cubic lattice composed of Na, Ln, and F has eight coordinated holes, which can contribute to nonradiative quenching through reduced migration of the exciton energy. (*Chem. Mater.* **26**, 1881-1888 (2014)) Therefore, the hexagonal-phased lattices in NaLuF_4 may hold more efficient energy transfer and thereby result in sufficient recombination energy that can be transferred to the doped Tb^{3+} , followed by the generation of persistent luminescence from the trivalent lanthanide.

The difference in persistent luminescence properties between cubic and hexagonal lattices has also received attention from other researchers. Tang *et al.* reported a similar phenomenon that only the excitation through the hexagonal-phased CsCdCl_3 host could give persistent emission, while the persistent emission cannot be observed for the cubic-phased CsCdCl_3 host. (*Angew. Chem. Int. Ed.* **61**, e202210975 (2022)) They also attributed the difference in persistent luminescence properties to the efficient energy transfer in hexagonal lattices, matching well with our findings and inference.

Otherwise, the difference in lattice types may result in different trap properties. Generally, the persistent performance of materials is closely related to the thermally-stimulated gradual release of charge carriers which are immobilized in the trap centers. (*J. Mater. Chem. C* **5**, 2844-2851 (2017)) The persistent properties are highly determined by the trap properties, which can be investigated by the analysis of the thermoluminescence (TL) spectra. (*Phys. Rev. B* **87**, 045126 (2013)) The TL intensity reflects the charge carrier concentration captured at the trap. (*J. Mater. Chem. C* **5**, 2844-2851 (2017); *Laser Photonics Rev.* **14**, 2000060 (2020).) We also investigate the TL behaviors of $\text{Na}_5\text{Lu}_9\text{F}_{32}:\text{Tb}^{3+}$ and $\text{NaLuF}_4:\text{Tb}^{3+}$ after the X-ray irradiation. The TL spectra were acquired on a self-assembled system comprising high-precision thermal stages (THMS600, British Linkam Scientific Instruments) and an Andor SR-500i spectrometer (Andor Technology Co. Belfast, UK) at a fixed heating rate of $3\text{ }^\circ\text{C s}^{-1}$ between room temperature and $350\text{ }^\circ\text{C}$. As shown in Supplementary Fig. 19, the TL spectrum of $\text{NaLuF}_4:\text{Tb}^{3+}$ displays a strong peak centered at $187\text{ }^\circ\text{C}$, while $\text{Na}_5\text{Lu}_9\text{F}_{32}:\text{Tb}^{3+}$ can not show any TL behavior. The results suggest that the traps in $\text{NaLuF}_4:\text{Tb}^{3+}$ exhibit much higher binding ability to charge carrier than that of $\text{Na}_5\text{Lu}_9\text{F}_{32}:\text{Tb}^{3+}$, which may also contribute to the difference between the persistent radioluminescence from $\text{NaLuF}_4:\text{Tb}^{3+}$ and synchronous radioluminescence behavior from $\text{Na}_5\text{Lu}_9\text{F}_{32}:\text{Tb}^{3+}$.

Supplementary Figure 19. Thermoluminescence (TL) spectra of $\text{Na}_5\text{Lu}_9\text{F}_{32}:\text{Tb}^{3+}$ and $\text{NaLuF}_4:\text{Tb}^{3+}$ measured after exposure with X-ray.

Finally, we still want to emphasize again that the above-mentioned analyses are based on the current models. The mechanism of the X-ray-induced persistent luminescence phenomenon is still under investigation and opening to question.

The above-mentioned results and discussion have been added in revised supplementary information.

2. Related Scheme 2 and Figure 3, how deep could HNTs@ $\text{Na}_5\text{Lu}_9\text{F}_{32}:\text{Tb}^{3+}$ penetrate into PUF? Is it only "a uniform coating of HNTs@ $\text{Na}_5\text{Lu}_9\text{F}_{32}:\text{Tb}^{3+}$ on the PUF surface"? Experimental data should be provided, e.g. using cross-section SEM images and elemental mapping.

Response: This is a nice suggestion. We have performed additional experiments and SEM mapping studies to reveal the uniformity of the composite foam and the penetration of the obtained HNTs@ $\text{Na}_5\text{Lu}_9\text{F}_{32}:\text{Tb}^{3+}$. For reviewer's convenience, we have added the details in the following:

A dried piece of 8.0 cm × 8.0 cm × 8.0 cm PUF was treated with HNTs@ $\text{Na}_5\text{Lu}_9\text{F}_{32}:\text{Tb}^{3+}$ to give the composite foam HNTs@ $\text{Na}_5\text{Lu}_9\text{F}_{32}:\text{Tb}^{3+}$ @PUF. The increased weight is calculated as 112%. Then the obtained composite foam was split in the middle, perpendicular to the bottom face. Five areas from bottom to up (0 cm, 2 cm, 4 cm, 6 cm, and 8 cm) were selected along the central axis in the cross-section and further analyzed by the SEM observation and elemental mapping studies (Supplementary Fig. 21).

The results indicate the distribution of Si, Al, Na, F, and Lu at the pore walls for

all cases. Because of the relatively low content of Tb in the obtained HNTs@Na₅Lu₉F₃₂:Tb³⁺, the distribution of Tb is difficult to achieve. The similar phenomenon can also be found in Fig. 3m in the main text. Si, Al, and F are the three most abundant elements in the EDS results (Supplementary Table 4) and further quantitatively analyzed to evaluate the uniformity among the five areas from bottom to up (0 cm, 2 cm, 4 cm, 6 cm, and 8 cm). The Atom % of Al is detected as 15.82, 15.48, 14.79, 14.39, and 15.22%, respectively, with a relative standard deviation (RSD) of 3.72%. The Atom % of Si is measured as 20.62, 22.10, 21.33, 21.69, and 21.73 %, respectively, with an RSD of 2.60%. The Atom % of F is 44.44, 44.03, 46.41, 47.43, and 44.74%, respectively, with an RSD of 3.19%. All of the results suggest that the HNTs@Na₅Lu₉F₃₂:Tb³⁺ shows a uniform distribution inside the 8.0 cm × 8.0 cm × 8.0 cm foam and a good penetration ability is demonstrated.

Supplementary Figure 21. Investigation of the cross-section micromorphology of Na₅Lu₉F₃₂:Tb³⁺-anchored halloysite nanotubes coated polyurethane foam (HNTs@Na₅Lu₉F₃₂:Tb³⁺@PUF). **a** Illustration of the sampling methods. **b** SEM images. **c-g** Elemental mapping images (**c** Si, **d** Al, **e** Na, **f** F, and **g** Lu).

Supplementary Table 4. Summary and analysis of the atom percentage of Al, Si, and F in the cross-section of Na₅Lu₉F₃₂:Tb³⁺-anchored halloysite nanotubes coated polyurethane foam (HNTs@Na₅Lu₉F₃₂:Tb³⁺@PUF) (8.0 cm × 8.0 cm × 8.0 cm) from SEM-EDS.

	Atom % - Al	Atom % - Si	Atom % - F
0 cm	15.82	20.62	44.44
2 cm	15.48	22.10	44.03
4 cm	14.79	21.33	46.41
6 cm	14.39	21.69	47.43
8 cm	15.22	21.73	44.74
Average	15.14	21.49	45.41
SD	0.56	0.56	1.45

RSD	3.72	2.60	3.19
-----	------	------	------

The above-mentioned results and discussion have been added in revised supplementary information.

3. The authors need to take measurements to get key parameters of this new type of X-ray scintillators and compare them with what have been reported in the literature. To keep readers better informed, it is suggested to compile a table and include in the ESI.

Response: This is a good suggestion. Because the X-ray scintillators reported in this study is a nanocomposite that can not form single crystals, some parameters used to evaluate conventional X-ray scintillators are inapplicable for HNTs@Na₅Lu₉F₃₂:Tb³⁺. We think we can address the reviewer's concern in an alternative way. The main purpose of this study is to develop X-ray scintillators that are water-dispersible, compatible with polymeric matrices, and readily processable to flexible substrates and hydrogels. Therefore, we made the comparisons between HNTs@Na₅Lu₉F₃₂:Tb³⁺ with non-perovskite-type X-ray scintillators that have been reported by summarizing the following items: application fields, materials types, and processing method. Additionally, the light yield and X-ray-excited emission wavelength (λ_{em}) were also included, the former serves as a main evaluation index of X-ray scintillators (*Angew. Chem. Int. Ed.* **134**, e202208440 (2022)). It can be observed from Supplementary Table 1 that the processing of X-ray scintillators into macroscopic materials, especially polymer composites, is still difficult without the use of organic solvent, high temperatures, or harsh conditions for crystal growth in literature. The obtained HNTs@Na₅Lu₉F₃₂:Tb³⁺ in our study is easily processable as aqueous dispersion to develop composite foams, flexible/hard screens, and hydrogels. Additionally, the light yield of the obtained HNTs@Na₅Lu₉F₃₂:Tb³⁺ was estimated to be 12300 photons MeV⁻¹, which is higher than well-known commercial scintillators including BaF₂ (1400 photons MeV⁻¹), Bi₄Ge₃O₁₂ (BGO, 8500 photons MeV⁻¹), and Gd₂SiO₅ (GSO):Ce (7000 photons MeV⁻¹). Though some single crystals, such as LYSO:Ce and LaBr₃:Ce, exhibit higher light yields, the high fabrication cost, harsh growth conditions, and non-flexibility limit their application to conventional hard devices.

On the other hand, biocompatibility is another important factor related to the application in biomedical field. Some X-ray scintillators are still suffering from the toxicity issues. To further demonstrate the advantage in biocompatibility, we performed additional cell viability tests by using mouse fibroblast L929 cells. The cell viability of mouse fibroblast L929 cells remained above 90% even at relatively high concentrations (1000 $\mu\text{g mL}^{-1}$) after 24 h and 48 h of treatment by HNTs@Na₅Lu₉F₃₂:Tb³⁺ (Supplementary Fig. 20). The results indicate that the obtained HNTs@Na₅Lu₉F₃₂:Tb³⁺ exhibits negligible biological toxicity on mouse fibroblast L929 cells.

Taking all aspects into consideration, the advantages of the obtained HNTs@Na₅Lu₉F₃₂:Tb³⁺ are summarized in the following:

i) The good water-dispersibility and desirable compatibility with polymer matrices

enable diverse aqueous processing approaches of radioluminescent foams, X-ray scintillating screens, and information encrypting hydrogels.

ii) The large length-diameter ratios can improve the mechanical properties of HNTs@Na₅Lu₉F₃₂:Tb³⁺-incorporated flexible X-ray scintillator screens.

iii) The light yield of 12300 photons MeV⁻¹ can be obtained under room temperature, which is higher than well-known commercial scintillators including BaF₂, BGO, and GSO:Ce.

iv) Low cytotoxicity

v) No need for harsh crystal growth conditions or high temperatures.

vi) Low cost of the raw materials and processing approaches.

vii) Good stability to heat and good light stability.

Supplementary Table 1. Comparison between the obtained Na₅Lu₉F₃₂:Tb³⁺-anchored halloysite nanotubes (HNTs@Na₅Lu₉F₃₂:Tb³⁺) with non-perovskite-type X-ray scintillators that have been reported in literature.

Sample	λ_{em}	Light yield (photons MeV ⁻¹)	Application	Material type	Processing method
Al(PO ₃) ₃ -CsPO ₃ -CsBr-CeBr ₃ glass	360 nm	2700	X-ray detection	Glass	Melted at 950 °C
BaF ₂ Crystals	350 and 425 nm	1400	Radiation absorption	Polystyrene composite film	Evaporation (C ₂ H ₄ Cl ₂ /CCl ₄)
BaF ₂ :Y Crystals	300 nm	2000	X-ray imaging	/	/
Bi ₄ Ge ₃ O ₁₂ (BGO) Crystals	480 nm	8500	X-ray detector	Hard film	Rapid annealing (750~800°C)
CsGd ₂ F ₇ :Ce nano-glass	380 nm	827	X-ray imaging	Hard film	Heat at 710 or 720 °C
Cu ₄ I ₄ : Tb MOFs	494, 547, 587, and 622 nm	29379	X-ray imaging	Flexible film	Evaporation (chloroform)
Gd ₂ SiO ₅ (GSO):Ce crystals	450 nm	7000	X-ray imaging	Hard film	Annealing (400~1000°C)
LaBr ₃ :Ce single crystal	380 nm	60000	Imager and spectrometer	Single crystal	Bridgman systems
LuF ₃ :Nd crystals	180, 230, and 250 nm	1200	Ultraviolet light emitting	Thin film	Laser deposition at 400 °C
Lu ₂ Si ₂ O ₇ :Nd single crystal	800~900 nm	800	/	/	/
Lu ₂ Si ₂ O ₇ :Pr single crystal	300~340 nm	9700	/	/	/
LYSO:Ce single crystal	420 nm	29000	X-ray imaging	Single crystal	Czochralski method
K ₃ Gd(PO ₄) ₂ :Ce crystals	335 and 360 nm	10217	X-ray detection	Crystals	Calcined at 1223 K
KLaF ₄ :Ce@KLa _{1-x} Gd _x F ₄ composite	380 nm	853	X-ray dosimetry	Fiber	Heating at 650 °C
PVK:Bi composite	440 nm	5600	X-ray detection	Plastic piece	Evaporation (THF)
Sr ₃ NbGa ₅ Si ₂ O ₁₄ crystals	400 nm	850	/	/	/
Y ₃ Al ₅ O ₁₂ (YAG):Ce single crystal	550 nm	17000	X-ray imaging	Single crystal	Czochralski method
ZnO single crystal	390 nm	9000	X-ray imaging	Nanowire field emitter	Electron beam evaporation (500°C)
ZnO:Ga single crystal	389 nm	714	Diagnose the distribution of cathode electron emission	Single crystal	Hydro-thermal method
Na ₅ Lu ₉ F ₃₂ :Tb ³⁺ crystals*	489, 544, 585, and 620 nm	15800	/	/	/

HNTs@Na ₅ Lu ₉ F ₃₂ :Tb ³⁺ nanotubes*	489, 544, 585, and 620 nm	12300	X-ray detection	PUF composite foam	Assembling in water under RT ^a
			X-ray imaging	Hard film	Evaporation (water)
			X-ray imaging	Flexible film	In-situ crosslinking in water
			Information encryption	Hydrogel	In-situ crosslinking in water

* Reported in this study

^a "RT" is abbreviated from room temperature

Supplementary Figure 20. Cell viability data on Na₅Lu₉F₃₂:Tb³⁺-anchored halloysite nanotubes (HNTs@Na₅Lu₉F₃₂:Tb³⁺) in mouse fibroblast L929 cells.

The above-mentioned results and discussion have been added in revised supplementary information. The first paragraph in Introduction Part is also revised accordingly.

To simply the response letter, citations were not included in the Table. The citations can be found the Supplementary Information.

4. What are the reasons/mechanisms for "some luminescent materials can exhibit different emission behavior between X-ray and UV light"? Are there any prior reports on this type of luminescent materials? If so, please list them in a table in ESI with adequate citations. And what are the advantages of the multilayer hydrogels reported in this manuscript?

Response: This is a good comment. The sentence "some luminescent materials can exhibit different emission behavior between X-ray and UV light" is used to emphasize the different excitation–emission mechanisms between UV-induced

photoluminescence (PL) and X-ray-induced radioluminescence (RL) processes in organic luminescent materials to explain the emission behavior from Gel-2. Organic materials composed of light atoms usually exhibit weak X-ray absorption (*Adv. Mater.* **30**, 1706956 (2018); *Nat. Photonics* **15**, 187-192 (2021)). On the other hand, X-ray beams holds much higher energy than UV/Vis light. A large number of triplet excitons can be generated with exposure to X-ray, which results in an intrinsic and huge loss channel (*Nat. Mater.* **21**, 210-216 (2022)). Therefore, many conventional organic luminescent materials can only generate luminescence under UV light and cannot emit light under X-ray irradiation. Take 1,4-phenyldiboronic for an example, it can be used as a crosslinker with PVA to prepare UV luminescent hydrogels, which have been reported in our previous study (*J. Colloid Interface Sci.* **617**, 353–362 (2022)), while it cannot provide spectroscopic capabilities under X-ray excitation.

As for inorganic crystal scintillators, the excitation–emission mechanisms are also remarkably different between UV/Vis-induced PL and X-ray-induced RL processes. UV/Vis excitation process can directly excite the luminescent center, whereas the X-ray excitation process interacts with holes and electrons from the host matrix (*Dalton Trans.* **47**, 13939 (2018)). The excitation that ultimately leads to RL should be due to an excitation of an electron from the valence band to the conduction band, while this process is difficult to take place in PL (*Appl. Phys. Lett.* **115**, 181104 (2019); *Nature* **561**, 88-93 (2018)). In the mechanism of PL, the excited valence electrons generated by optical excitation will return to the ground state, accompanied by emitting photons. The dopant ions can be directly excited by incident photon energy in this process. As a contrast, in the RL case, the dopants are indirectly excited: X-ray irradiation firstly generates photoelectrons or Compton electrons; the energy carried by the electrons then excites the dopant ions. Consequently, energy loss may take place in the indirectly RL process and give rise to different emission behaviors (*Mater. Lett.* **144**, 43–45(2015)).

Some cases in prior reports have been summarized in Supplementary Table 3, which may be helpful for potential readers interested in this topic. The above-mentioned results and discussion have been added in revised supplementary information. For review’s convenience, we have also added the Supplementary Table 3 in the following.

To simply the response letter, citations were not included in the Table. The citations can be found the Supplementary Information.

Supplementary Table 3. Summary of the materials exhibit different emission behaviors between UV/Vis-induced photoluminescence and X-ray-induced radioluminescence.

Sample	UV/Vis-induced photoluminescence	X-ray-induced radioluminescence
LaB ₃ O ₆ :Ce crystals	$\lambda_{em} = 303$ nm ($\lambda_{ex} = 277$ nm)	$\lambda_{em} = 323$ nm
Y ₃ TaO ₇ :Tm crystals	$\lambda_{em} = 460, 471, 487, 610, 632,$ and 655 nm	$\lambda_{em} = 355, 392, 455, 472, 487, 515, 658,$ and 797 nm
NaMgF ₃ :Sm nanoparticles	$\lambda_{em} = 450$ (broad), $557, 593, 611, 639,$ and 698 nm ($\lambda_{ex} = 280$ nm)	$\lambda_{em} = 555, 599, 636, 694, 723,$ and 800 nm
CaF ₂ :Th crystals	Distinct peaks between 260 and 500 nm	$\lambda_{em} = 230$ and 400 nm
KMgF ₃ :Eu nanoparticles	$\lambda_{em} = 360$ nm ($\lambda_{ex} = 290$ nm)	$\lambda_{em} = 540$ – 640 nm
La ₂ Hf ₂ O ₇ :Eu nanoparticles	$\lambda_{em} = 582, 612, 630,$ and 712 nm ($\lambda_{ex} = 258$ nm; λ_{max} locates at 612 or 630 nm)	Multi-peaks in the region 580 – 630 nm (λ_{max} occurs before 600 nm) and 700 – 725 nm

Cs ₃ BiCl ₆ crystals	$\lambda_{em} = 390$ nm ($\lambda_{ex} = 325$ nm)	In addition to the luminescence band at 390 nm, another band was observed at 600-700 nm
CdTa ₂ O ₆ :Eu crystals	$\lambda_{em} = 592, 612,$ and 623 nm ($\lambda_{ex} = 465.7$ nm)	$\lambda_{em} = 430, 592, 599, 619, 662, 713, 754,$ and 822 nm
Yb ²⁺ -doped silica glass	$\lambda_{em} = 506$ nm ($\lambda_{ex} = 399$ nm)	$\lambda_{em} = 525$ nm
BaSnO ₃ : Tb ceramic	$\lambda_{em} = 541$ and 897 nm ($\lambda_{ex} = 325$ nm)	$\lambda_{em} = 492, 541, 583, 619,$ and 897 nm
NaMgF ₃ : Mn nanoparticles	$\lambda_{em} = 498$ and 602 nm ($\lambda_{ex} = 396$ nm)	$\lambda_{em} = 602$ nm
(PEA) ₂ MnCl ₄ crystals	$\lambda_{em} = 604$ nm ($\lambda_{ex} = 366$ nm)	Non-emission at room temperature
(PPA) ₂ MnCl ₄ crystals	$\lambda_{em} = 581$ nm ($\lambda_{ex} = 366$ nm)	Non-emission at room temperature

The advantages of the multilayer hydrogels reported in this manuscript were summarized in the following and have been added in revised manuscript:

To the best of our knowledge, there is no report on hydrogels for information encryption by using X-ray as the decoding tool. Besides this, the multilayer hydrogels support an extra safety encryption technology to prevent information leakage and combat fakes, because the “false information” can be read from multilayer hydrogels under UV light that serves as a commonly-used decoding tool in conventional anti-counterfeiting and encryption technologies. The multilayer hydrogels are assembled based on the dynamic covalent bonds rather than physically attached, which can afford uniform hydrogels and further allows more complicated programming for information camouflage and multilevel encryption based on the three different types component hydrogels: non-emission (Gel-0), PL (Gel-2) and PL&RL (Gel-1). The presence of dynamic covalent bonds in the obtained multilayer hydrogels would also contribute to potential self-healing properties.

5. To keep readers well informed, the authors are suggested to provide citations related to "Some other organic materials can only generate luminescence under UV light and cannot emit light under X-ray irradiation."

Response: We have added some explanations and citations on this phenomenon in revised manuscript. For review’s convenience, we have also added them in the following:

“Some organic materials composed of light atoms exhibit weak X-ray absorption, which can only generate luminescence under UV light and cannot emit light under X-ray irradiation. (*Nat. Commun.* **13**, 3995 (2022); *Adv. Mater.* **30**, 1706956 (2018); *Nat. Photonics* **15**, 187-192 (2021))”

6. The authors mentioned that "The toxicity of lead is another severe issue restricting wide spread applications of perovskite-based X-ray scintillators" in the Introduction. However, many reported studies on perovskite-based X-ray scintillators do not contain lead, for example, Ref. 8,9 and 11 used lead-free perovskites. The authors should compare their results with what have been reported before this manuscript could be validated for publication on Nature Communications.

Response: This is a good comment. We find it is necessary to provide more explanation to clarify the necessity to develop non-perovskite-type X-ray scintillators. Besides the lead-caused concerns on toxicity, the X-ray luminescence of perovskite-based X-ray scintillators is usually affected by the strong thermal quenching effects even under room temperature because of the low band gaps. Take CsPbBr₃ for an example, it holds an high light yield of ~50000 photons MeV⁻¹ at 7K while the light yield is determined as <500 photons MeV⁻¹ under room temperature (*Sci. Rep.* **10**, 8601 (2020); *Nucl. Instrum. Methods. Phys. Res. B* **592**, 369-373 (2008)). Though some organic-inorganic perovskites have demonstrated excellent performance in optoelectronic devices, the serious thermal quenching effects still restrict their applications of X-ray scintillators. CH₃NH₃PbI₃ and CH₃NH₃PbBr₃ exhibit a high light yield of ~150000 photons MeV⁻¹ at 10K, however, the high yields are measured as <1000 photons MeV⁻¹ under room temperature (*Sci. Rep.* **6**, 37254 (2016)). It should be noted that the thermal quenching effect is not limited in lead-containing perovskites. A light yield of ≈110000 photons MeV⁻¹ can be obtained for Rb₂AgBr₃, while the high yield dramatically decreases as the increased temperature (*Adv. Funct. Mater.* **31**, 2007921 (2021)). Though the light yield of PEA₂MnCl₄ is expected to be 200000 photons MeV⁻¹ based on the theoretical calculations, the absence of X-ray RL signals was observed for PEA₂MnCl₄ (*J. Phys. D Appl. Phys.* **53**, 455303 (2020)). The same quenching phenomenon is also observed for PPA₂MnCl₄ (*J. Phys. D Appl. Phys.* **53**, 455303 (2020)). Up to know, the thermal quenching effect in perovskite-type X-ray scintillator is still under investigation and difficult to be predicted. On the other hand, it is also difficult to get rid of the use of organic solvents, high temperatures, or harsh conditions for crystal growth in practical use. Some single crystal perovskites with light yield are still suffering from the difficulty in processing, high fabrication cost, harsh growth conditions, and non-flexibility. Therefore, in this study, we focus on anchoring non-perovskite-type scintillators on HNTs.

For reviewer's convenience, we have also added Supplementary Table 2 in the following. It contains the comparisons between HNTs@Na₅Lu₉F₃₂:Tb³⁺ with some perovskite-based X-ray scintillators that have been reported in literature. To simply the response letter, citations were not included in the Table. The citations can be found the Supplementary Information.

Supplementary Table 2. Comparison between the obtained Na₅Lu₉F₃₂:Tb³⁺-anchored halloysite nanotubes (HNTs@Na₅Lu₉F₃₂:Tb³⁺) with perovskite-based X-ray scintillators that have been reported in literature.

Sample	λ_{em}	Light yield (photons MeV ⁻¹)	Application	Material type	Processing method
CaHfO ₃ :Ce single crystal	430 nm	7800	/	/	/
Cs ₂ BaBr ₄ :Tl crystals	380 and 510 nm	1700~2700	/	/	/
Cs ₃ BiCl ₆ crystals	390 and 600~700 nm	800	/	/	/
Cs ₂ HfCl ₆ :Te crystals	575 nm	9000~13100	/	/	/
CsCaCl ₃ crystals	250 and 305 nm	410	/	/	/
CsCaCl ₃ :Ce crystals	375 nm	7600	/	/	/
Cs ₃ Cu ₂ I ₅ crystals	570 nm	31700	X-ray imaging	Hard film	390 °C heating in nitrogen-filled glovebox
Cs ₃ Cu ₂ I ₅ nanocrystals	445 nm	79279	X-ray imaging	Thick film	Evaporation (hexane)
CsPbBr ₃ crystals	533, 548, and 573 nm	50000 (7 K ^a) ¹⁸ < 500 (RT ^b) ¹⁹	X-ray detection	Flexible film	Evaporation (dichloromethane)
CsPbBr ₃ nanosheets	525 nm	21000	X-ray imaging	Hard film	Assembly in toluene
CsPbBr ₃ :Eu quantum dots	595, 616, 654, and 701 nm	10100	X-ray imaging	Glass-ceramic	500 °C heating
CsPbBr ₃ :F nanocrystals	/	8500	X-ray detection	Flexible film	Evaporation (dichloromethane)
CsPbBr ₃ :Lu nanocrystals	516 nm	/	X-ray imaging	Hard film	Melt (1200 °C)-annealed (420 °C)-heat (500 °C)
CsPbBr ₃ @BaF ₂ composite	Tunable from 435 to 648 nm	6300	X-ray detection	Solid screen	Evaporation (dodecane/ethyl acetate)
CsPbCl ₃ single crystal	415 nm	330	/	/	/
Cs ₂ SnF ₆ :Mn nanocrystals	600-650 nm	3000	X-ray imaging	Flexible film	Evaporation (dichloromethane)
CsSrCl ₃ :Ce crystals	350-400 nm	8600	/	/	/
Cs ₂ ZnBr ₄ :Mn crystals	526 nm	15600	X-ray imaging	Flexible film	Evaporation (ethanol)
Cs ₂ ZnCl ₄ nanorod	254, 305, 400, 500, and 760 nm	100-300	/	/	/
Rb ₂ AgBr ₃ crystals	480 nm	110000 (80K)	X-ray imaging	Thick film	550 °C heating
TlCdCl ₃ crystal	450 nm	2200	/	/	/
(A) ₂ (MA) _{n-1} Pb _n Br _{3n+1} nanocrystals	542 nm	/	X-ray imaging	Flexible film	Spin-coating using chlorobenzene in nitrogen-filled glovebox
(BM) ₂ PbBr ₄ crystals	480 nm	3190	/	/	/
(CH ₃ NH ₃)PbI ₃ crystals	770 nm	150000 (10 K) < 1000 (RT)	/	/	/
(CH ₃ NH ₃)PbBr ₃ crystals	540 nm	150000 (10 K) < 1000 (RT)	/	/	/
(C ₆ H ₅ C ₂ H ₄ NH ₃) ₂ PbBr ₄ single crystal	410 nm	14000	Radioactive element detection	Single crystal	Evaporation (DMF)
(C ₆ H ₅ C ₂ H ₄ NH ₃) ₂ PbI ₄ single crystal	560	2900	/	/	/
(C ₁₃ H ₁₄ N ₃) ₃ SbCl ₆ crystals	/	2000	/	/	/

$((C_{38}H_{34}P_2)MnBr_4)$ single crystal	517 nm	~ 80,000	X-ray imaging	Flexible film	Grind into powder and dispersed in Organosilicone
$(C_{24}H_{20}P)_2MnBr_4$ nanocrystals	520 nm	/	X-ray imaging	Flexible film	Evaporation (DMF)
(EDBE) $PbCl_4$ crystals	520 nm	1200000 (130 K) 9000 (RT)	/	/	/
$(PEA)_2MnCl_4$ crystals	604 nm	200000 (DFT ^c) ~0 (RT)	/	/	/
$(PEA)_2PbBr_4 \cdot Li$ single crystal	450 to 750 nm	11000	X-ray imaging	Hard film	Evaporation (DMSO)
$(PEA)_2PbI_4$ crystals	532 and 660 nm	10000 (10 K) 1000 (RT)	/	/	/
(S-3AP) $PbBr_3Cl \cdot H_2O$ single crystal	575 nm	19000	X-ray imaging	Single crystal	Grow from HCl&HBr
$Na_5Lu_9F_{32} \cdot Tb^{3+}$ crystals*	489, 544, 585, and 620 nm	15800 (RT)	/	/	/
$HNTs@Na_5Lu_9F_{32} \cdot Tb^{3+}$ nanotubes*	489, 544, 585, and 620 nm	12300 (RT)	X-ray detection	PUF composite foam	Assembling in water under RT
			X-ray imaging	Hard film	Evaporation (water)
			X-ray imaging	Flexible film	In-situ crosslinking in water
			Information encryption	Hydrogel	In-situ crosslinking in water

* Reported in this study;

^a “K” is abbreviated from Kelvin;

^b “RT” is abbreviated from room temperature;

^c “DFT” is abbreviated from density functional theory.

The above-mentioned results and discussion have been added in revised Supplementary Information. The first paragraph in Introduction Part is also revised accordingly.

7. By the way, the title reads too long. It should be shortened significantly to signify the key message of the work.

Response: This is a nice suggestion. The title has been revised as “Water-dispersible X-ray scintillators enabling radioluminescent foams, X-ray imaging, and information encrypting hydrogels”.

8. There are typos and grammatical errors to be corrected. For example,

Line 100, page 4: "In contrast tot the persistent radioluminescence"

Line 331, page 12: "the here developed lead-free composite"

Response: Thanks for your carefully check. We have revised the sentences as “In contrast to the persistent radioluminescence” and “the developed lead-free composite”, respectively. In addition, we also have carefully polished the whole manuscript according to the “formatting instructions”.

Other revisions

Additionally, we have performed some other revisions on format according to the “formatting instructions”. To simplify the text, the changes relating to format issues were not marked. Here, we briefly summarize the revisions in the following:

1. Graphical Abstract is deleted because it is not allowed in “Nat. Commun.”
2. Abstract is shortened to 150 words. Title is also shortened to less than 15 words.
3. Subheading numbers in the main text have been deleted.
4. “cGy/s” is revised as “cGy s⁻¹”. Congeneric cases have also been revised accordingly.
5. The order of the affiliations in title page is revised according to the rule “List affiliations in the same numerical order as their first appearance in the author list”.
6. Panels in figures have been revised by using the “a, b, c...”. Citations in the texts are also revised accordingly. “Supplementary Figure 1” is used to replace “Figure S1”. Congeneric cases have also been revised accordingly.
7. Reference list has been polished according to the rules in “formatting instruction”.
8. The last two paragraphs in Introduction Part have been combined into one paragraph. Following this way, the final paragraph begins with “In this work”.
9. Scheme 4 and Supplementary Figure 16 have been redrawn to improve the quality.
10. Exaggerated language, like “extremely”, has been revised or deleted.
11. Fig. 2a is redrawn to avoid using red and green simultaneously.
12. Abbreviations are defined in each figure/table legend.
13. Other revisions in format and grammar.

REVIEWERS' COMMENTS

Reviewer #5 (Remarks to the Author):

The manuscript has been revised thoroughly to convincingly address the raised points. I recommend its publication in its present form.